# Optimal Regret for Policy Optimization in Contextual Bandits

**Orin Levy** [1]  **Yishay Mansour** [1][2]

## Abstract

We present the first high-probability optimal regret bound for a policy optimization technique applied to the problem of stochastic contextual multi-armed bandit (CMAB) with general offline function approximation. Our algorithm is both efficient and achieves an optimal regret bound of $\widetilde{O}(\sqrt{K|\mathcal{A}|\log|\mathcal{F}|})$, where $K$ is the number of rounds, $\mathcal{A}$ is the set of arms, and $\mathcal{F}$ is the function class used to approximate the losses. Our results bridge the gap between theory and practice, demonstrating that the widely used policy optimization methods for the contextual bandit problem can achieve a rigorously-proved optimal regret bound. We support our theoretical results with an empirical evaluation of our algorithm.

## 1. Introduction

*Policy Optimization* (PO) methods are among the most practical techniques in Reinforcement Learning (RL), with impressive empirical success across a wide range of tasks (Schulman et al., 2017). The applications of policy optimization span various domains, from recommendations (Jain et al., 2025) to training robots (Peters and Schaal, 2006; 2008; Levine et al., 2016; Gu et al., 2017) and control tasks (Mnih et al., 2015; Schulman et al., 2016; Lillicrap et al., 2016), to its notable successes in Large Language Models (LLMs) fine-tuning (Stiennon et al., 2020; Glaese et al., 2022; Ouyang et al., 2022). Motivated by the great success of context-based policy optimization methods in aligning LLMs with human preferences (Rafailov et al., 2023), we revisit the widely studied model of *Contextual Multi-Armed Bandits* (CMABs; Auer et al. (2002)). CMABs naturally model many real-world online tasks where external factors influence the outcome of a chosen strategy. This includes online advertisement and recommendation systems, where user preferences affect whether they click on the proposed item or not; healthcare, where a patient's medical history impacts their reaction to a treatment, and more.

The CMAB problem describes an online decision-making scenario, where external factors affect the decision. We refer to these factors as the *context*. In each round $k$ of $K$-rounds game, the agent first observes a new context $c_k$ selected from a huge context space $\mathcal{C}$. Given the current context, the agent chooses an action $a_k$ from a finite set of actions $\mathcal{A}$, and suffers a loss $\ell_k$ associated with the context $c_k$ and the action $a_k$. We emphasize that the context determines the loss for each action, meaning that for different contexts, the optimal action-selection strategy might be completely different.

The agent aims to minimize the cumulative loss suffered throughout a $K$-round game. Since the action-selection strategy in CMAB is context-dependent, we consider it as a contextual *policy*. We measure the performance of the agent in terms of *regret*, which is the cumulative loss of the agent when compared to the cumulative loss of the best contextual policy. Hence, the agent's goal is to minimize regret.

Due to its high relevance, CMAB has been extensively studied, both theoretically and empirically. On the theoretical side, CMAB has been studied under several assumptions and different learning setups, which we will elaborate on later. On the empirical side, the work of Bietti et al. (2021) is the most notable.

In this paper, we consider *stochastic* CMAB, in which the context in each round is sampled from an unknown distribution. In this setting, the context space is typically very large, so the agent is likely to never observe the same context twice. Since the optimal action depends on the context, achieving sublinear regret is impossible without further assumptions. To address this, the stochastic CMAB literature has focused on the *offline function approximation* framework; a minimal and realistic setting in which the desired regret rate of $\widetilde{O}(\sqrt{K})$ becomes achievable.

In this framework, the agent is provided with a *realizable* and finite loss function class $\mathcal{F}$, where each function maps context-action pairs to an expected loss value. Realizability means that the true loss function is in the class $\mathcal{F}$. Moreover, the agent does not have direct access to the function class,

[1]School of Computer Science and AI, Tel-Aviv University, Tel-Aviv, Israel [2]Google Research, Tel-Aviv, Israel. Correspondence to: Orin Levy <orinlevy@mail.tau.ac.il>, Yishay Mansour <mansour.yishay@gmail.com>.

*Proceedings of the 43rd International Conference on Machine Learning*, Seoul, South Korea. PMLR 306, 2026. Copyright 2026 by the author(s).

but instead interacts with it through an offline regression oracle, which returns a candidate function that best fits the dataset of observed losses at each round.

Under these assumptions, state-of-the-art algorithms (Foster et al., 2021b; Simchi-Levi and Xu, 2022; Xu and Zeevi, 2020) achieve an optimal regret bound of $\widetilde{O}(\sqrt{K|\mathcal{A}|\log|\mathcal{F}|})$ efficiently, assuming access to an efficient offline regression oracle. UCCB algorithm of Xu and Zeevi (2020) constructs confidence bounds and use a deterministic action selection rule that plays the best optimistic arm in each round. In contrast, (Simchi-Levi and Xu, 2022; Foster et al., 2021b) introduce inverse gap weighting (IGW)-based techniques, that define a stochastic policy where the probability of selecting each action is proportional to its estimated suboptimality under the current loss predictor. They prove, respectively, optimal worst-case and instance-dependent regret bounds. However, despite the success of PO methods in other domains, the existing stochastic CMAB literature lacks a rigorously provable efficient algorithm that leverages PO within the offline function approximation framework to minimize regret.

PO methods are widely used in real-world applications, as discussed earlier, owing to their closed-form and interpretable update rules that balance exploration and exploitation without the need to solve additional optimization problems. Their exponential-weighting updates yield smoother and more stable policy adjustments than IGW or deterministic UCB-based methods, since they moderate the sensitivity of the policy update to changes in the loss estimator. Since for CMAB, these estimators are obtained from least-squares regression that can be highly sensitive to new samples, methods like IGW and UCCB often exhibit sharp fluctuations: IGW must rely on a rather involved low-switching mechanism to control such instability, while UCCB commits to an optimistic deterministic policy that forfeits the inherent exploration benefits of a stochastic policy. In contrast, PO's stochastic and gradual updates naturally improve robustness to estimator variability, without requiring explicit low-switching scheduling mechanisms (Simchi-Levi and Xu, 2022; Foster et al., 2021b). For all these reasons, we believe that developing a rigorously-proved variant of PO for regret minimization in stochastic CMAB with offline function approximation-based loss estimators is an important and currently unaddressed gap in the literature, which we aim to close.

**Summary of our main contributions.** Our primary goal is to develop a rigorously provable policy optimization method for regret minimization in stochastic CMABs that uses loss estimators constructed via offline function approximation. To this end, we introduce OPO-CMAB, a new policy optimization-based algorithm for stochastic CMABs with general offline function approximation. Our algorithm is

computationally efficient (assuming access to an efficient regression oracle) and achieves an optimal regret bound of $\widetilde{O}(\sqrt{K|\mathcal{A}|\log|\mathcal{F}|})$, which holds with high probability.

On the technical side, we address the challenge of ensuring sufficient exploration when the agent plays a PO stochastic policy that relies on offline function approximation-based loss estimators. To obtain this, we generalize the counterfactual confidence bounds of Xu and Zeevi (2020) to stochastic policies and derive computable counterfactual exploration bonuses that integrate into the policy-optimization update rule. To the best of our knowledge, this is the first work to achieve such guarantees without making additional assumptions regarding the function class (e.g., Eluder dimension (Levy et al., 2024)).

On the empirical side, we implemented our algorithm and evaluated its performance on the VowpalWabbit benchmark suite (Bietti et al., 2021), demonstrating its competitiveness compared to SOTA baselines.

Our results show that policy optimization can be adapted to the stochastic CMAB setting with general function approximation, achieving both provable optimal regret and competitive empirical performance.

## 1.1. Related Literature Review

**Contextual Multi-Armed Bandits (CMAB).** The study of CMABs has gained significant attention in the last decade, with various assumptions made about the contexts (i.e., adversarially or stochastically chosen), the function classes used (i.e., policy or loss/rewards classes), and the oracles used, if any. Existing literature can be categorized to two main streams.

The first stream focuses on learning a close-to-best policy within specific given finite policy class $\Pi$. It began with the well-known EXP4 algorithm for adversarial CMAB (Auer et al., 2002), followed by Dudik et al. (2011) and Agarwal et al. (2014), who explored computationally efficient methods for stochastic CMAB. These two works proved an optimal regret bound of $\widetilde{O}(\sqrt{K|\mathcal{A}|\log|\Pi|})$.

The second stream, which is also the stream this work belongs to, pertains to the realizable function approximation setting. In which, the agent has access to a realizable rewards or losses function class (used to approximate the rewards/losses for each context and action), which she accesses via an optimization oracle. Langford and Zhang (2007) considered stochastic contexts and obtained suboptimal regret. Later, Agarwal et al. (2012) presented a regressor elimination algorithm and obtained a regret bound of $\widetilde{O}(\sqrt{K|\mathcal{A}|\log|\mathcal{F}|})$, where $\mathcal{F}$ represents a finite set of realizable contextual reward functions used to approximate the reward. The downside of this algorithm is that the runtime complexity of the algorithm scales with

$|\mathcal{F}|$. They also presented a matching lower bound of $\Omega(\sqrt{K|\mathcal{A}|\log|\mathcal{F}|/\log|\mathcal{A}|})$ proves the optimality of their upper bound. Foster et al. (2018a) initiated the research of regression oracle based efficient algorithms for stochastic CMAB. Finally, Simchi-Levi and Xu (2022) use inverse gap weighting (IGW) technique to derive optimal worst-case regret and Foster et al. (2021b) extends them to obtain instance-dependent regret bound. Xu and Zeevi (2020) use upper confidence bounds for deterministic policies to also derive optimal worst-case regret. The previously mentioned algorithms access the function class using a standard offline least-squares regression oracle, and derive algorithms that obtain optimal regret and can be implemented efficiently, where the efficiency is depending on the run-time complexity of the oracle in use. We, in contrast, apply a policy optimization technique over an optimistic loss approximation, computed for stochastic polices. Similarly, our algorithm is efficient, assuming an efficient offline regression oracle, and obtains optimal regret bounds.

Research has also considered adversarially chosen contexts, notably through the works of (Foster and Rakhlin, 2020; Foster and Krishnamurthy, 2021; Foster et al., 2021a; Zhu et al., 2022), who introduced IGW-based policies using online regression oracles for accessing the function class $\mathcal{F}$. These efforts resulted in an optimal regret bound of $\widetilde{O}(\sqrt{K|\mathcal{A}|\mathcal{R}_K(\mathcal{O})})$, where $\mathcal{R}_K(\mathcal{O})$ denotes the regret of the oracle used.

An additional related model is linear CMAB, with Abe and Long (1999) being the first to consider this model, and the current state-of-the-art regret minimization algorithms were introduced by Abbasi-Yadkori et al. (2011) and Chu et al. (2011). Our framework is more general than linear function approximation.

**PO Methods in Reinforcement Learning (RL).** The theoretical analysis of PO techniques has been extensively studied, mostly in the basic RL setup of tabular Markov Decision Processes (MDPs), an RL environment with finite state and action spaces, dynamics, and rewards or losses associated with each state-action pair. Even-Dar et al. (2009) initiated the theoretical research of policy optimization methods in RL by proposing and analyzing the weighted-majority algorithm for MDPs in the setup of known dynamics, adversarial losses, and full feedback. Later, Neu et al. (2010) extended their technique to handle bandit feedback. Shani et al. (2020) presented the optimistic policy optimization algorithm and analyzed it in the case of unknown dynamics and bandit feedback while considering both stochastic and adversarial losses. For stochastic losses, they obtained rate-optimal regret; however, in the case of adversarial losses, their regret bound was sub-optimal. Luo et al. (2021) improved their result by applying policy optimization with refined exploration bonuses and obtained rate-optimal regret in the

challenging case of unknown dynamics, adversarial losses, and bandit feedback.

Policy optimization has been applied to more complex setups in tabular MDPs, such as aggregated feedback (Lancewicki and Mansour, 2025), delayed feedback (Lancewicki et al., 2022; 2023), and many other setups with various assumptions.

Beyond the tabular setting, policy optimization has been applied to linear MDPs. Cai et al. (2020) obtained optimal-rate regret in the unknown dynamics and adversarial losses model, assuming full feedback. Luo et al. (2021) also presented a $\widetilde{O}(K^{2/3})$ regret for the linear case, assuming access to a simulator, where $K$ is the number of episodes. Later, Dai et al. (2023) improved the regret to $\widetilde{O}(\sqrt{K})$ under the same assumptions. Sherman et al. (2023) obtained a $\widetilde{O}(K^{6/7})$ regret bound efficiently, without assuming access to a simulator, which was later improved by Liu et al. (2024) to $\widetilde{O}(K^{4/5})$. Recently, Sherman et al. (2024) obtained a rate-optimal regret of $\widetilde{O}(\sqrt{K})$ using a reward-free based warm-up. Later, Cassel and Rosenberg (2024) obtained similar regret bound without warm-up.

To the best of our knowledge, pure policy optimization has not been studied in RL literature beyond tabular and linear MDPs. Our work is the first attempt to use policy optimization as an exploration method on top of general function approximation for CMABs.

## 2. Preliminaries and Notations

We consider the problem of *stochastic Contextual Multi-Armed Bandit (CMAB)*, in which we have a finite discrete set of arms $\mathcal{A}$, containing $|\mathcal{A}|$ arms. In the online learning scenario, in each round $k$ of $K$ rounds game, a fresh context $c \in \mathcal{C}$ is sampled from an unknown distribution $\mathcal{D}$ over $\mathcal{C}$. In general, the context space $\mathcal{C}$ can be infinite, but, for mathematical convenience, we assume it is huge but finite[1]. For each context $c \in \mathcal{C}$ and action $a \in \mathcal{A}$ there is an associated stochastic loss with expectation $\ell(c, a) = \mathbb{E}[L(c, a)|c, a]$, where $L(c, a) \in [0, 1]$. After observing the context, the agent selects an action to play and suffers the related stochastic loss.

We refer to the agent's action selection strategy as a *(stochastic) contextual policy* $\pi : \mathcal{C} \to \Delta(\mathcal{A})$ that maps contexts to a distribution over actions. We refer to $\pi(c, a)$ as the probability dictated by $\pi$ to play action $a$ given the context is $c$. The *optimal policy* $\pi_\star$ satisfies for each $c \in \mathcal{C}$ that $\pi_\star(c, \cdot) \in \arg\min_{p \in \Delta(\mathcal{A})} \langle p, \ell(c, \cdot) \rangle$, where $\langle \cdot, \cdot \rangle$ denotes inner product between these two vectors. The interaction

---

[1]This assumption is standard in CMAB literature, as the extension to infinite context space is straightforward by applying appropriate discretization. The goal is to obtain a regret bound that is independent of the cardinality of $\mathcal{C}$.

protocol with stochastic CMAB is as follows. In each round $k = 1, 2, \ldots, K$: (1) A fresh context $c_k$ is sampled from $\mathcal{D}$; (2) The agent selects action $a_k \sim \pi_k(c_k, \cdot)$ to play; (3) The agent suffer loss $\ell_k = L(c_k, a_k)$.

**Learning objective.** We measure the performance of our algorithm in terms of the *(pseudo) regret* in comparison to the optimal policy $\pi_\star$, which is formally defined as $\mathcal{R}_K := \sum_{k=1}^{K} \langle \pi_k(c_k, \cdot) - \pi_\star(c_k, \cdot), \ell(c_k, \cdot) \rangle$, where $\langle \cdot, \cdot \rangle$ denotes inner product. The expected (pseudo) regret is then $\mathbb{E}\mathcal{R}_K := \sum_{k=1}^{K} \mathbb{E}_{c_k} [\langle \pi_k(c_k, \cdot) - \pi_\star(c_k, \cdot), \ell(c_k, \cdot) \rangle]$, where the expectation is over the contexts sampled throughout the game. Our goal is to develop an efficient, policy optimization-based regret minimization algorithm for stochastic CMAB.

**Additional notations.** Throughout the paper, we denote by $|x|$ the absoulote value of any $x \in \mathbb{R}$. Also, for any two distributions $p, q$ over (finite) support $\mathcal{X}$ we denote by $d_{KL}(p||q)$ the Kullback-Leibler (KL) divergence between $p$ and $q$ that is defined as $d_{KL}(p||q) := \sum_{x \in \mathcal{X}} p(x) \log \frac{p(x)}{q(x)}$.

### 2.1. Offline Function Approximation

As previous literature for stochastic CMAB (e.g., (Simchi-Levi and Xu, 2022; Xu and Zeevi, 2020; Agarwal et al., 2012; Langford and Zhang, 2007)) shows, an offline function approximation is necessary to obtain non-trivial regret for stochastic CMAB. Hence, in compatible with previous works, we assume access to a realizable and finite[2] losses function class $\mathcal{F} \subseteq \mathcal{C} \times \mathcal{A} \to [0, 1]$ via an offline least-squares regression oracle denoted $\mathcal{O}_{sq}^{\mathcal{F}}$.

**Assumption 2.1** (Realizability). There exist $f_\star \in \mathcal{F}$ such that $\forall (c, a) \in \mathcal{C} \times \mathcal{A}$ it holds that $f_\star(c, a) = \ell(c, a)$.

The function class is being accessed via an offline least-squares regression oracle, next defined.

**Assumption 2.2** (Offline least-squares regression oracle). Given a dataset $D_n = \{(c_i, a_i, \ell_i)\}_{i=1}^{n}$ we assume access to an offline oracle $\mathcal{O}_{sq}^{\mathcal{F}}$ that returns a candidate solution $\hat{f}_{n+1}$ to the following Empirical Risk Minimization (ERM) problem with respect to the square loss: $\hat{f}_{n+1} \in \arg\min_{f \in \mathcal{F}} \sum_{i=1}^{n} (f(c_i, a_i) - \ell_i)^2$.

Access to an offline least-square regression oracle is also a commonly used assumption in stochastic CMAB literature (see, e.g., (Simchi-Levi and Xu, 2022; Xu and Zeevi, 2020)). The reason behind the choice of least-squares regression for the loss approximation is that least-squares regression is both compatible with approximating an expectation given

---

[2]The assumption is standard as the extension to infinite function classes is also straightforward, see (Shalev-Shwartz and Ben-David, 2014) for offline regression with infinite function classes. We consider finite function classes for the sake of mathematical convenience and readability.

stochastic examples and can be implemented efficiently for many function classes, with the clear example of a linear functions, for which the solution is given by a closed form.

## 3. Algorithm and Main Result

`OPO-CMAB`, described in Algorithm 1, is an optimistic policy optimization algorithm for stochastic CMABs.

An integral ingredient of our algorithm is the generalization of the *counterfactual exploration bonuses* (Xu and Zeevi, 2020) to stochastic policies. Our exploration bonus for arm $a$ at round $k$ and context $c$ is

$$b_k^{\beta}(c, a) = \min\left\{1, \frac{\beta/2}{1 + \sum_{i=1}^{k-1} \pi_i(c, a)}\right\}, \qquad (1)$$

where $\beta = O(\sqrt{K})$ is a tunable parameter controlling the overall exploration level.

The intuition behind these bonuses is that they quantify how well explored each arm is for the current context. To see this, consider a counterfactual scenario in which the same context $c$ had appeared in all previous rounds. At each past round $i = 1, \ldots, k - 1$, the agent would have sampled arms according to the past policies $\pi_1(c, \cdot), \pi_2(c, \cdot), \ldots, \pi_{k-1}(c, \cdot)$. Hence, the imaginary expected number of times arm $a$ would have been played for this context is $\mathbb{E}\left[\sum_{i=1}^{k-1} \pi_i(c, a) \mid c, a\right]$. Thus, this counterfactual quantity serves as a realization of the expected number of times arm $a$ would have been chosen for $c$ up to round $k$. When this quantity is large, the algorithm has implicitly gathered substantial information about arm $a$ for the context $c$, and thus assigns it a small bonus. Conversely, when it is small, the bonus remains large, encouraging further exploration. When balanced by $\beta$, this bonus is $\approx \widetilde{O}(1/\sqrt{k})$, similar to the standard exploration bonus in stochastic MABs (Slivkins et al., 2019).

Based on this optimistic exploration principle, our algorithm performs policy optimization by applying exponential policy improvements, as is standard in RL literature (see, e.g., (Shani et al., 2020)). Given the first observed context $c_1$, the agent samples an action uniformly at random from $\pi_1$, plays it, and updates the oracle with the resulting observation. Then, for rounds $t = 2, 3, \ldots, K$, the agent computes the policies as described next. She observes the current context $c_t$ and counterfactually computes all past policies $\pi_1(c_t, \cdot), \ldots, \pi_{t-1}(c_t, \cdot)$ in order to compute the exploration bonus and subsequently derive $\pi_t(c_t, \cdot)$. For each $k = 1, 2, \ldots, t - 1$, to compute the next policy $\pi_{k+1}(c_t, \cdot)$, the agent uses the approximated loss returned by the oracle at round $k$ (computed from the data collected in rounds $\{1, \ldots, k - 1\}$) denoted by $\hat{f}_k$. From this approximation, she subtracts the counterfactual exploration bonus from Equation (1), which depends on the probabilities of all past

policies $\pi_1, \ldots, \pi_{k-1}$ for playing each specific action $a$ under the current context $c_t$. The next policy is then defined by an exponential improvement of the current policy with respect to its optimistic loss approximation.

---

**Algorithm 1** Optimistic Policy Optimization for CMAB (`OPO-CMAB`)

---

1: **Inputs:** learning rate $\eta$, number of episodes $K$, exploration parameter $\beta$.
2: **Initialization:** $\pi_1$ is the uniform distribution over actions for all $c \in \mathcal{C}$; $f_1 \in \mathcal{F}$ is chosen arbitrarily.
3: **for** $t = 1, 2, \ldots, K$ **do**
4:     Observe fresh context $c_t$.
5:     **if** $t \geq 2$ **then**
6:         **for** $k = 1, \ldots, t-1$ **do**
7:             Compute for all $a \in \mathcal{A}$:
            $\hat{\ell}_k(c_t, a) = \max\{0, \hat{f}_k(c_t, a) - b_k^\beta(c_t, a)\}$
            where $b_k^\beta(c, a) = \min\left\{1, \frac{\beta/2}{1 + \sum_{i=1}^{k-1} \pi_i(c, a)}\right\}$.
            {Policy Evaluation}
8:             Compute for all $a \in \mathcal{A}$:
            $\pi_{k+1}(c_t, a) = \frac{\pi_k(c_t, a) \exp\left(-\eta \hat{\ell}_k(c_t, a)\right)}{\sum_{a' \in \mathcal{A}} \pi_k(c_t, a') \exp\left(-\eta \hat{\ell}_k(c_t, a')\right)}$
            {Policy Improvement}
9:         **end for**
10:     **end if**
11:     Sample action $a_t \sim \pi_t(c_t, \cdot)$; play $a_t$; observe $\ell_t$.
12:     Update the loss approximation using the optimization oracle $\mathcal{O}_{\text{sq}}^{\mathcal{F}}$:

$$\hat{f}_{t+1} \in \arg\min_{f \in \mathcal{F}} \sum_{i=1}^{t} (f(c_i, a_i) - \ell_i)^2$$

13: **end for**

---

The next theorem states the regret bound of `OPO-CMAB`.

**Theorem 3.1** (Regret bound). *For any $\delta \in (0, 1)$ let $\beta = \sqrt{34K \log(4|\mathcal{F}|K^3/\delta)/|\mathcal{A}|}$ and $\eta = \sqrt{\log|\mathcal{A}|/K}$. Then, with probability at least $1 - \delta$,*

$$\mathcal{R}_K \leq \widetilde{O}\left(\sqrt{K|\mathcal{A}|\log(|\mathcal{F}|/\delta)}\right).$$

**Discussion.** Our algorithm integrates a policy optimization update rule with offline function approximation for loss prediction. We employ an optimistic approximation by incorporating exploration bonuses tailored for both offline function approximation and stochastic policies. Intuitively, the bonus assigned to each action reflects the counterfactual expected number of samples that would have been collected for that action if the current context had been observed throughout all previous rounds. The loss estimators are clipped to $[0, 1]$ to ensure that they are bounded. The algorithm is computationally efficient with $O(K^2)$ run-time complexity, assuming access to an efficient oracle.

## 4. Regret Analysis

In this section, we analyze the regret of Algorithm 1, proving Theorem 3.1. We follow the regret decomposition of Shani et al. (2020) but adapt it to function approximation in the model of stochastic CMAB. We start our analysis by noting that with probability at least $1 - \delta/2$, $\mathcal{R}_K \leq \mathbb{E}\mathcal{R}_K + 2\sqrt{2K \log(4/\delta)}$. As the contexts are iid, and the policies $\{\pi_k\}_{k=1}^{K}$ are determined completely by the history, the above is a direct implication of Azuma-Hoeffding's inequality. (Full proof is given in Corollary B.3).

Hence, we focus on bounding the expected regret, which will imply a high probability regret bound, by the above. To obtain the desired bound, we decompose the expected regret as follows.

$$\mathbb{E}\mathcal{R}_K = \sum_{k=1}^{K} \mathbb{E}_{c_k}\left[\left\langle \pi_k(c_k, \cdot), \ell(c_k, \cdot) - \hat{\ell}_k(c_k, \cdot)\right\rangle\right] \quad (2)$$

$$+ \sum_{k=1}^{K} \mathbb{E}_{c_k}\left[\left\langle \pi_k(c_k, \cdot) - \pi_\star(c_k, \cdot), \hat{\ell}_k(c_k, \cdot)\right\rangle\right] \quad (3)$$

$$+ \sum_{k=1}^{K} \mathbb{E}_{c_k}\left[\left\langle \pi_\star(c_k, \cdot), \hat{\ell}_k(c_k, \cdot) - \ell(c_k, \cdot)\right\rangle\right] \quad (4)$$

where term (2) stands for the expected approximation error of the loss with respect to the played policies $\{\pi_k\}_{k=1}^{K}$, term (3) is the sub-optimality of the true optimal policy $\pi_\star$ on the approximated loss function in each round, and term (4) is the expected approximation error of the loss with respect to the optimal policy. In what follows, we bound each of the terms separately.

We start our analysis by stating a uniform convergence guarantee over the sequence of regressors computed throughout the run. Lemma 5 in Xu and Zeevi (2020) (see Lemma B.4) presents such guarantee with respect to any function sequence. An immediate corollary of it implies a uniform convergence guarantee of an offline least squares regression oracle. The corollary is below, for proof see Corollary B.5.

**Corollary 4.1** (uniform convergence of offline least-squares regression). *Let $\hat{f}_2, \hat{f}_3, \ldots \in \mathcal{F}$ denote the sequence of least squares minimizers and let $\pi_1, \pi_2, \ldots$ denote the sequence of contextual played policies. The following holds for any $\delta \in (0, 1)$ and $t \geq 2$ with probability at least $1 - \delta/4$.*

$$\sum_{i=1}^{t-1} \mathbb{E}_{c_i}\left[\mathbb{E}_{a_i \sim \pi_i(c_i, \cdot)}\left[\left(\hat{f}_t(c_i, a_i) - f_\star(c_i, a_i)\right)^2\right]\right]$$
$$\leq 68 \log(4|\mathcal{F}|t^3/\delta).$$

Next, we observe that the sum of bonuses is upper-bounded, in expectation over the context.

**Lemma 4.2** (Bonuses bound). *The following holds.*

$$\sum_{k=1}^{K} \mathbb{E}_{c_k}\left[\sum_{a\in\mathcal{A}} \pi_k(c_k, a) b_k^{\beta}(c_k, a)\right] \le \beta|\mathcal{A}|\log(K+1).$$

This bound implied by applying an algebraic calculation yields a logarithmic upper bound over the sum. See Lemma A.1 for more details. Using both bounds above, we derive the following upper bound of term (2).

**Lemma 4.3** (Term (2) bound). *For any $\delta \in (0, 1)$, let $\beta = \sqrt{34K\log(4|\mathcal{F}|K^3/\delta)/|\mathcal{A}|}$. Then, with probability at least $1 - \delta/4$, it holds that*

$$\sum_{k=1}^{K} \mathbb{E}_{c_k}\left[\left\langle \pi_k(c_k, \cdot), \ell(c_k, \cdot) - \hat{\ell}_k(c_k, \cdot)\right\rangle\right]$$
$$\le \tilde{O}\left(\sqrt{K|\mathcal{A}|\log(|\mathcal{F}|/\delta)}\right).$$

Below we provide a proof sketch; the complete proof appears in Lemma A.2.

*Proof sketch.* We begin our proof by observing that

$$(2) \le \sum_{k=1}^{K} \mathbb{E}_{c_k}\left[\left\langle \pi_k(c_k, \cdot), f_\star(c_k, \cdot) - \hat{f}_k(c_k, \cdot)\right\rangle\right] \quad (5)$$

$$+ \sum_{k=1}^{K} \mathbb{E}_{c_k}\left[\sum_{a\in\mathcal{A}} \pi_k(c_k, a) b_k^{\beta}(c_k, a)\right]. \quad (6)$$

Hence, we focus in upper bounding (5). Since all functions in $\mathcal{F}$ are bounded in $[0, 1]$, and by absolute value properties, we have that (5) is upper bounded by

$$1 + \sum_{k=2}^{K} \mathbb{E}_{c_k}\left[\sum_{a\in\mathcal{A}} \pi_k(c_k, a)\min\{1, |f_\star(c_k, a) - \hat{f}_k(c_k, a)|\}\right].$$

Henceforth, we are neglecting the $+1$ term. We then multiply the absolute value, which is the second argument of the $\min\{\cdot, \cdot\}$, by $\sqrt{\frac{\beta}{\beta}\frac{1+\sum_{i=1}^{k-1}\pi_i(c_k, a)}{1+\sum_{i=1}^{k-1}\pi_i(c_k, a)}}$ and apply AM-GM inequality to upper bound the latter by

$$\sum_{k=2}^{K} \mathbb{E}_{c_k}\left[\sum_{a\in\mathcal{A}} \pi_k(c_k, a)\min\left\{1, \frac{\beta/2}{1+\sum_{i=1}^{k-1}\pi_i(c_k, a)}\right.\right.$$
$$\left.\left. + \frac{1}{2\beta}\left(1+\sum_{i=1}^{k-1}\pi_i(c_k, a)\right)\left(f_\star(c_k, a) - \hat{f}_k(c_k, a)\right)^2\right\}\right].$$

By minimum properties and reordering, we obtain the latter is upper bounded by

$$\sum_{k=1}^{K} \mathbb{E}_{c_k}\left[\sum_{a\in\mathcal{A}} \pi_k(c_k, a) b_k^{\beta}(c_k, a)\right] + \frac{K}{2\beta}$$

$$+ \frac{1}{2\beta}\sum_{k=2}^{K}\sum_{i=1}^{k-1}\mathbb{E}_{c_i}\left[\mathbb{E}_{a\sim\pi_i(c_i, \cdot)}\left[\left(f_\star(c_i, a) - \hat{f}_k(c_i, a)\right)^2\right]\right],$$

where the first term will be summed with (6), and together they are bounded using Lemma 4.2. The last term is bounded with a probability of at least $1 - \delta/4$, by the oracle convergence guarantees of Corollary 4.1. Summing all together, we obtain that with a probability of at least $1 - \delta/4$,

$$(2) \le 2\beta|\mathcal{A}|\log(K+1) + \frac{K}{2\beta} + \frac{68\log(4|\mathcal{F}|K^3/\delta)K}{2\beta}.$$

By setting $\beta$ as specified, we derive the lemma. $\square$

The upper bound on term (3) follows directly from the standard analysis of Online Mirror Descent (OMD; see Appendix B.1). For full proof see Lemma A.3.

**Lemma 4.4** (Term (3) bound). *For the choice in $\eta = \sqrt{\log|\mathcal{A}|/K}$, the following holds true.*

$$\sum_{k=1}^{K} \mathbb{E}_{c_k}\left[\left\langle \pi_k(c_k, \cdot) - \pi_\star(c_k, \cdot), \hat{\ell}_k(c_k, \cdot)\right\rangle\right]$$
$$\le O(\sqrt{K\log|\mathcal{A}|}).$$

*Proof sketch.* We observe that in each round of Algorithm 1, the policy we compute for each observed context $c \in \mathcal{C}$ separately, is the solution of the exactly same optimization problem of OMD algorithm, using the KL divergence for the Bregman's divergence term. As our loss estimators are bounded in $[0, 1]$ and $\pi_1(c, \cdot)$ is uniform over the actions for every context $c$, we can apply the fundamental inequality of online mirror descent (see Theorem B.1), to obtain the following for each fixed context $c$ separately.

$$\sum_{k=1}^{K}\left\langle \hat{\ell}_k(c, \cdot), \pi_k(c, \cdot) - \pi_\star(c, \cdot)\right\rangle \le \frac{\log|\mathcal{A}|}{\eta} + \frac{\eta K}{2}.$$

For $\eta = \sqrt{2\log|\mathcal{A}|/K}$ we obtain for each context $c$ that this sum is upper bounded by $O(\sqrt{K\log|\mathcal{A}|})$, which implies the lemma. $\square$

Lastly, we bound term (4) using similar proof technique to that of term (2). See Lemma A.4 for proof.

**Lemma 4.5** (Term (4) bound). *For any $\delta \in (0, 1)$ let $\beta = \sqrt{34K\log(4|\mathcal{F}|K^3/\delta)/|\mathcal{A}|}$. Then, the following holds with probability at least $1 - \delta/4$.*

$$\sum_{k=1}^{K} \mathbb{E}_{c_k}\left[\left\langle \pi_\star(c_k, \cdot), \hat{\ell}_k(c_k, \cdot) - \ell(c_k, \cdot)\right\rangle\right]$$
$$\le \tilde{O}\left(\sqrt{K|\mathcal{A}|\log(|\mathcal{F}|/\delta)}\right).$$

Using all the above, we prove Theorem 3.1.

*Proof of Theorem 3.1.* By taking a union bound over the events specified in Lemmas 4.3 and 4.5, and combine it with the result of Lemma 4.4, we obtain that with probability at least $1 - \delta/2$, $\mathbb{E}\mathcal{R}_k \leq \widetilde{O}\left(\sqrt{K|\mathcal{A}|\log(|\mathcal{F}|/\delta)}\right)$, which implies that with probability at least $1 - \delta$, $\mathcal{R}_k \leq \widetilde{O}\left(\sqrt{K|\mathcal{A}|\log(|\mathcal{F}|/\delta)}\right)$. $\square$

# 5. Experiments

As is standard in the CMAB literature, we evaluate our algorithm using the VowpalWabbit (VW)[3] benchmark suite (Bietti et al., 2021), which implements the practical CMAB algorithms[4]. The state-of-the-art empirical comparison on this benchmark is by Foster and Krishnamurthy (2021), who extensively evaluated the efficient, regression oracle based CMAB algorithms (SquareCB, AdaCB, RegCB) and a supervised learning baseline Supervised against their proposed algorithm FastCB, across various hyper-parameters and both square and logistic losses. They found FastCB with logistic loss performed best, followed by SquareCB (logistic and squared loss), while AdaCB and RegCB lagged behind. Following this setup, we compare our method OPO-CMAB to FastCB, SquareCB, RegCB, AdaCB, and the supervised baseline Supervised.

**Implementation Details.** Following prior work, all algorithms are implemented in VW using an online regression oracle for both linear regression and logistic regression. We integrated OPO-CMAB into VW and implemented it as in Algorithm 1, with small changes: (i) the exploration bonus is set adaptively as $\beta_k = \gamma\sqrt{k/|\mathcal{A}|}$ with $\gamma$ being a tuned hyper-parameter; (ii) $\eta$ is now also a tuned hyper-parameter and (iii) support also logistic regression for better accommodating to multi-class and multi-label tasks as in the tested datasets.

**Experimental setup and evaluation.** Previous works (Foster and Krishnamurthy, 2021; Bietti et al., 2021) evaluate all algorithms on more than 500 multi-class and multi-label classification datasets from OpenML. Due to computational limitations, we evaluated all algorithms on 18 relatively small datasets selected among those. As these are multiclass classification datasets, we simulate bandit feedback as the agent receives loss 0 for a correct prediction and 1 otherwise, similarly to the prior works.

**Performance** is measured by the average loss, also known as the *Progressive Validation (PV)* loss (Blum et al., 1999), which for algorithm A and a dataset of $K$ examples defined

as $L_{PV}(\text{A}, K) = \frac{1}{K}\sum_{k=1}^{K}\ell_k(a_k)$. Hence, the PV-loss directly reflects regret differences between algorithms.

Following previous work, we tuned the hyper-parameters of each algorithm by choosing the configuration that achieves the lowest final PV-loss for each dataset. The set of tested values for each parameter can be found in Table 1. The parameters values for the known algorithms were chosen as described in previous work. For our OPO-CMAB, we select $\gamma$ and $\eta$ values from a small set of relevant values on a discretized 10-scale grid. The final parameter values chosen for each algorithm and dataset can be found in Table 2. We then run the selected configuration on 10 random permutations of the related dataset. In the presented plots, we compare the averaged PV-loss decay curve of each algorithm as well as Standard deviation (Std) on three selected datasets, in which the supervised baseline converged to non-trivial final PV-loss. In Appendix C, we support those plots by presenting a table summarizes the mean differences in the final PV-loss of each algorithm from the supervised baseline (see Figure 2), as well as provide more related details.

**Discussion of results in Figure 1.** For dataset 1084, all algorithms appear to reach a plateau after 100 examples, which is expected for a 3-armed CMAB instance. As anticipated, Supervised achieves the lowest PV-losses throughout the run, closely followed by OPO-CMAB, with RegCB slightly higher. In fourth place is FastCB. SquareCB ranks fifth, and AdaCB is last, both with a notable gap from the other algorithms.

For dataset 1062, it appears that all algorithms nearly reach a plateau fast, as expected for 2-armed CMAB. Supervised achieves the lowest PV-losses throughout the run, followed by FastCB and then OPO-CMAB, which have slightly higher PV loss values. In fourth place is AdaCB, then RegCB with higher PV-losses, and finally SquareCB.

For dataset 1015, all algorithms appear to reach a plateau after 50 examples, as expected for a 2-armed CMAB. AdaCB achieves the lowest PV-loss throughout the run, followed closely by FastCB, Supervised, and RegCB, which have very similar PV-loss values; SquareCB performs slightly worse. For this dataset, our OPO-CMAB shows slightly higher PV-loss values than the others, but its performance remains competitive.

In Appendix C, Figure 2, we present a difference-from-supervised comparison that exemplifies the same trend.

Overall, the experiments demonstrate that all algorithms exhibit similar performance on the tested datasets, with no significant evidence that any one algorithm outperforms the others. This supports our claim that our algorithm is competitive with previous CMAB algorithms. For more details see Appendix C.

---

[3]https://vowpalwabbit.org/

[4]Our code is publicly available at https://github.com/orinL/OPO_CMAB_Experiments_code.

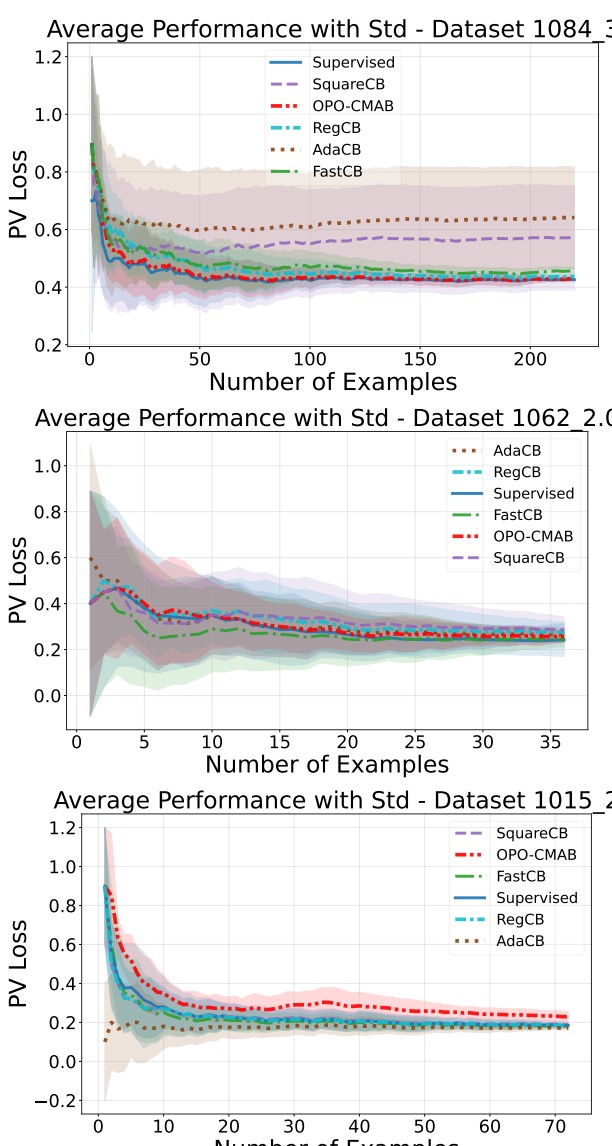

*Figure 1.* Averaged PV loss in Datasets 1084, 1062, 1015.

## 6. Conclusions and Discussion

In this paper we present provably efficient regret minimization algorithm for stochastic CMABs with general offline function approximation, based on policy optimization. The algorithm is computationally efficient (given an efficient oracle) and achieves an optimal high probability regret bound of $\widetilde{O}(\sqrt{K|\mathcal{A}|\log|\mathcal{F}|})$. Our results can be immediately extended to infinite contexts and infinite function classes using appropriate discretizations or learning dimensions (Shalev-Shwartz and Ben-David, 2014), which we left aside for clarity and readability. A natural direction for future work is to extend our theoretical results beyond CMABs to more general settings such as contextual RL, RL with large state spaces, and other rich RL problems. We hope our results will inspire such further research.

Empirically, our method demonstrates competitive performance across a range of multiclass datasets, with no significant evidence that any algorithm consistently outperforms the others. However, our implementation of `OPO-CMAB` in VW is limited by its $O(K^2)$ runtime complexity. Since our primary contribution is theoretical, we focused on demonstrating applicability and competitiveness, rather than developing a practical implementation. Thus, we did not explore optimizations for `OPO-CMAB`. Nevertheless, we believe our technique could inspire many efficient heuristics involving policy optimization updates with function approximation-based loss predictors. Potential directions include using noisy batching instead of exhaustive computation of exploration bonuses, heuristic look-back rather than recovering all history, and more. Developing such practical and scalable heuristics of our technique is a promising direction for future research.

## Impact Statement

This paper presents work whose goal is to advance the field of Machine Learning. There are many potential societal consequences of our work, none which we feel must be specifically highlighted here.

## Acknowledgments

OL thanks Alon Peled-Cohen for fruitful discussions and Idan Schwartz for his assistance with running the experiments.

This project has received funding from the European Research Council (ERC) under the European Union's Horizon 2020 research and innovation program (grant agreement No. 882396), by the Israel Science Foundation, the Yandex Initiative for Machine Learning at Tel Aviv University and a grant from the Tel Aviv University Center for AI and Data Science (TAD).

OL is also supported by the Google PhD Fellowship Award (2025).

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

# A. Proofs: Regret Analysis

In this section, we provide the full analysis of the regret of Algorithm 1, proving Theorem 3.1. We follow the regret analysis structure of Shani et al. (2020) but adapt it to function approximation for stochastic CMAB. We start our analysis by noting that with probability at least $1 - \delta/2$,

$$\mathcal{R}_K \leq \mathbb{E}\mathcal{R}_K + 2\sqrt{2K \log(4/\delta)}.$$

As the contexts are iid, and the policies $\{\pi_k\}_{k=1}^K$ are determined by the history, the above is a direct implication of Azuma-Hoeffding's inequality. (Full proof is given in Corollary B.3).

Hence, we focus on bounding the expected regret, which will imply a high probability regret bound, by the above. To obtain the desired bound, we decompose the expected regret as follows.

$$\mathbb{E}\mathcal{R}_K = \sum_{k=1}^K \mathbb{E}_{c_k}[\langle \pi_k(c_k, \cdot) - \pi_\star(c_k, \cdot), \ell(c_k, \cdot) \rangle]$$

$$= \underbrace{\sum_{k=1}^K \mathbb{E}_{c_k}\left[\langle \pi_k(c_k, \cdot), \ell(c_k, \cdot) \rangle - \left\langle \pi_k(c_k, \cdot), \hat{\ell}_k(c_k, \cdot) \right\rangle\right]}_{(2)}$$

$$+ \underbrace{\sum_{k=1}^K \mathbb{E}_{c_k}\left[\left\langle \pi_k(c_k, \cdot), \hat{\ell}_k(c_k, \cdot) \right\rangle - \left\langle \pi_\star(c_k, \cdot), \hat{\ell}_k(c_k, \cdot) \right\rangle\right]}_{(3)}$$

$$+ \underbrace{\sum_{k=1}^K \mathbb{E}_{c_k}\left[\left\langle \pi_\star(c_k, ), \hat{\ell}_k(c_k, \cdot) \right\rangle - \langle \pi_\star(c_k, \cdot), \ell(c_k, \cdot) \rangle\right]}_{(4)}$$

We bound each term separately, but first, we upper-bound the sum of bonuses.

**Lemma A.1** (Bonuses bound, restatement of Lemma 4.2). *The following holds.*

$$\sum_{k=1}^K \mathbb{E}_{c_k}\left[\sum_{a \in \mathcal{A}} \pi_k(c_k, a) b_k^\beta(c_k, a)\right] \leq \beta|\mathcal{A}| \log(K+1).$$

*Proof.* We first note that

$$\sum_{k=1}^K \mathbb{E}_{c_k}\left[\sum_{a \in \mathcal{A}} \pi_k(c_k, a) b_k^\beta(c_k, a)\right] = \sum_{k=1}^K \mathbb{E}_{c_k}\left[\sum_{a \in \mathcal{A}} \pi_k(c_k, a) \min\left\{1, \frac{\beta/2}{1 + \sum_{i=1}^{k-1} \pi_i(c_k, a)}\right\}\right]$$

$$\leq \frac{\beta}{2} \sum_{k=1}^K \mathbb{E}_{c_k}\left[\sum_{a \in \mathcal{A}} \frac{\pi_k(c_k, a)}{1 + \sum_{i=1}^{k-1} \pi_i(c_k, a)}\right]$$

$$= \frac{\beta}{2} \mathbb{E}_c\left[\sum_{a \in \mathcal{A}} \sum_{k=1}^K \frac{\pi_k(c, a)}{1 + \sum_{i=1}^{k-1} \pi_i(c, a)}\right].$$

Next, we fix a context $c \in \mathcal{C}$ and bound

$$\sum_{a \in \mathcal{A}} \sum_{k=1}^K \frac{\pi_k(c, a)}{1 + \sum_{i=1}^{k-1} \pi_i(c, a)}.$$

Using Lemma C.1 from Levy et al. (2024), given in Lemma B.6, we obtain that for any fixed context it holds that

$$\sum_{a \in \mathcal{A}} \sum_{k=1}^K \frac{\pi_k(c, a)}{1 + \sum_{i=1}^{k-1} \pi_i(c, a)} \leq 2|\mathcal{A}| \log(K+1).$$

By taking an expectation over the context on both sides, we obtain

$$\mathbb{E}_c\left[\sum_{a\in\mathcal{A}}\sum_{k=1}^{K}\frac{\pi_k(c,a)}{1+\sum_{i=1}^{k-1}\pi_i(c,a)}\right]\leq 2|\mathcal{A}|\log(K+1).$$

Lastly, we plug the latter into our first inequality and conclude the lemma. □

We then move to bound each one of the three terms.

**Lemma A.2** (Term (2) bound, restatement of Lemma 4.3)**.** *For any $\delta\in(0,1)$, let $\beta=\sqrt{\frac{34K\log(4|\mathcal{F}|K^3/\delta)}{|\mathcal{A}|}}$. Then, with probability at least $1-\delta/4$ it holds that*

$$\sum_{k=1}^{K}\mathbb{E}_{c_k}\left[\langle\pi_k(c_k,\cdot),\ell(c_k,\cdot)\rangle-\left\langle\pi_k(c_k,\cdot),\hat{\ell}_k(c_k,\cdot)\right\rangle\right]\leq\widetilde{O}\left(\sqrt{K|\mathcal{A}|\log(|\mathcal{F}|/\delta)}\right).$$

*Proof.* The following holds with probability at least $1-\delta/4$.

$$\sum_{k=1}^{K}\mathbb{E}_{c_k}\left[\langle\pi_k(c_k,\cdot),\ell(c_k,\cdot)\rangle-\left\langle\pi_k(c_k,\cdot),\hat{\ell}_k(c_k,\cdot)\right\rangle\right]$$

$$=\sum_{k=1}^{K}\mathbb{E}_{c_k}\left[\sum_{a\in\mathcal{A}}\pi_k(c_k,a)\left(\ell(c_k,a)-\max\left\{0,\hat{f}_k(c_k,a)-b_k^{\beta}(c_k,a)\right\}\right)\right]$$

$$\leq\sum_{k=1}^{K}\mathbb{E}_{c_k}\left[\sum_{a\in\mathcal{A}}\pi_k(c_k,a)\left(\ell(c_k,a)-\left(\hat{f}_k(c_k,a)-b_k^{\beta}(c_k,a)\right)\right)\right]$$

$$=\sum_{k=1}^{K}\mathbb{E}_{c_k}\left[\sum_{a\in\mathcal{A}}\pi_k(c_k,a)\left(f_\star(c_k,a)-\hat{f}_k(c_k,a)\right)\right]+\sum_{k=1}^{K}\mathbb{E}_{c_k}\left[\sum_{a\in\mathcal{A}}\pi_k(c_k,a)b_k^{\beta}(c_k,a)\right]$$

$$\leq\sum_{k=1}^{K}\mathbb{E}_{c_k}\left[\sum_{a\in\mathcal{A}}\pi_k(c_k,a)|f_\star(c_k,a)-\hat{f}_k(c_k,a)|\right]+\sum_{k=1}^{K}\mathbb{E}_{c_k}\left[\sum_{a\in\mathcal{A}}\pi_k(c_k,a)b_k^{\beta}(c_k,a)\right]$$

$$=\sum_{k=1}^{K}\mathbb{E}_{c_k}\left[\sum_{a\in\mathcal{A}}\pi_k(c_k,a)\min\left\{1,|f_\star(c_k,a)-\hat{f}_k(c_k,a)|\right\}\right]+\sum_{k=1}^{K}\mathbb{E}_{c_k}\left[\sum_{a\in\mathcal{A}}\pi_k(c_k,a)b_k^{\beta}(c_k,a)\right]$$

(Since both function are bounded in $[0,1]$, so is the absolute value)

$$\leq\sum_{k=2}^{K}\mathbb{E}_{c_k}\left[\sum_{a\in\mathcal{A}}\pi_k(c_k,a)\min\left\{1,|f_\star(c_k,a)-\hat{f}_k(c_k,a)|\right\}\right]+\sum_{k=1}^{K}\mathbb{E}_{c_k}\left[\sum_{a\in\mathcal{A}}\pi_k(c_k,a)b_k^{\beta}(c_k,a)\right]+1$$

$$=\sum_{k=2}^{K}\mathbb{E}_{c_k}\left[\sum_{a\in\mathcal{A}}\pi_k(c_k,a)\min\left\{1,\sqrt{\frac{\beta}{\beta}\frac{1+\sum_{i=1}^{k-1}\pi_i(c_k,a)}{1+\sum_{i=1}^{k-1}\pi_i(c_k,a)}}|f_\star(c_k,a)-\hat{f}_k(c_k,a)|\right\}\right]$$

$$+\sum_{k=1}^{K}\mathbb{E}_{c_k}\left[\sum_{a\in\mathcal{A}}\pi_k(c_k,a)b_k^{\beta}(c_k,a)\right]+1$$

$$\leq\sum_{k=2}^{K}\mathbb{E}_{c_k}\left[\sum_{a\in\mathcal{A}}\pi_k(c_k,a)\min\left\{1,\frac{1}{2}\left(\frac{\beta}{1+\sum_{i=1}^{k-1}\pi_i(c_k,a)}+\frac{1}{\beta}\left(1+\sum_{i=1}^{k-1}\pi_i(c_k,a)\right)\left(f_\star(c_k,a)-\hat{f}_k(c_k,a)\right)^2\right)\right\}\right]$$

(By AM-GM inequality)

$$+\sum_{k=1}^{K}\mathbb{E}_{c_k}\left[\sum_{a\in\mathcal{A}}\pi_k(c_k,a)b_k^{\beta}(c_k,a)\right]+1$$

$$\leq \sum_{k=2}^{K} \mathbb{E}_{c_k}\left[\sum_{a\in\mathcal{A}} \pi_k(c_k,a)\min\left\{1, \frac{\beta/2}{1+\sum_{i=1}^{k-1}\pi_i(c_k,a)}\right\}\right] + \frac{K}{2\beta}$$

(Since all terms in the $\min$ are positive for $\beta > 0$, holds by $\min\{\cdot,\cdot\}$ properties)

$$+\frac{1}{2\beta}\sum_{k=2}^{K}\mathbb{E}_{c_k}\left[\sum_{i=1}^{k-1}\mathbb{E}_{a\sim\pi_i(c_k,\cdot)}\left[\left(f_\star(c_k,a)-\hat{f}_k(c_k,a)\right)^2\right]\right] + \sum_{k=1}^{K}\mathbb{E}_{c_k}\left[\sum_{a\in\mathcal{A}}\pi_k(c_k,a)b_k^\beta(c_k,a)\right] + 1$$

$$\leq 2\sum_{k=1}^{K}\mathbb{E}_{c_k}\left[\sum_{a\in\mathcal{A}}\pi_k(c_k,a)b_k^\beta(c_k,a)\right] + \frac{K}{2\beta} + \frac{1}{2\beta}\sum_{k=2}^{K}\sum_{i=1}^{k-1}\mathbb{E}_{c_i}\left[\mathbb{E}_{a\sim\pi_i(c_i,\cdot)}\left[\left(f_\star(c_i,a)-\hat{f}_k(c_i,a)\right)^2\right]\right] + 1$$

(By the bonus definition in Algorithm 1, Lemma B.4 assumptions and linearity of expectation, as the context is iid.)

$$\leq 2\sum_{k=1}^{K}\mathbb{E}_{c_k}\left[\sum_{a\in\mathcal{A}}\pi_k(c_k,a)b_k^\beta(c_k,a)\right] + \frac{K}{2\beta} + \frac{68\log(4|\mathcal{F}|K^3/\delta)K}{2\beta} + 1$$

(Holds with probability at least $1-\delta/4$, by Corollary 4.1)

$$\leq 2\beta|\mathcal{A}|\log(K+1) + \frac{K}{2\beta} + \frac{68\log(4|\mathcal{F}|K^3/\delta)K}{2\beta} + 1.$$

(By Lemma A.1)

Finally, by setting $\beta = \sqrt{\frac{34K\log(4|\mathcal{F}|K^3/\delta)}{|\mathcal{A}|}}$ we obtain that term (2) is bounded as

$$\sum_{k=1}^{K}\mathbb{E}_{c_k}\left[\langle\pi_k(c_k,\cdot),\ell(c_k,\cdot)\rangle - \left\langle\pi_k(c_k,\cdot),\hat{\ell}_k(c_k,\cdot)\right\rangle\right] \leq \widetilde{O}\left(\sqrt{K|\mathcal{A}|\log(|\mathcal{F}|/\delta)}\right).$$

$\square$

We continue to bound term (3).

**Lemma A.3** (Term (3) bound, restatement of Lemma 4.4). *For the choice in* $\eta = \sqrt{\log|\mathcal{A}|/K}$*, the following holds true.*

$$\sum_{k=1}^{K}\mathbb{E}_{c_k}\left[\left\langle\pi_k(c_k,\cdot) - \pi_\star(c_k,\cdot),\hat{\ell}_k(c_k,\cdot)\right\rangle\right] \leq O(\sqrt{K\log|\mathcal{A}|}).$$

*Proof.* Observe that the policy we compute in Algorithm 1 for each observed context $c \in \mathcal{C}$ is the solution of the same optimization problem as in Online Mirror Descent (OMD) algorithm (see Appendix B.1 for more information), using the KL-divergence for the Bregman divergence term.

Formally, our policy satisfies the following for every context $c \in \mathcal{C}$ and round $k \in [K]$.

$$\pi_{k+1}(c,\cdot) \in \arg\min_{\pi\in\Delta(\mathcal{A})}\eta\left\langle\hat{\ell}_k(c,\cdot),\pi - \pi_k(c,\cdot)\right\rangle + d_{KL}(\pi||\pi_k(c,\cdot)). \tag{7}$$

Hence, term (3) is the linear approximation term of the policy computed by the OMD algorithm, in expectation over the contexts.

As our loss estimators are bounded in $[0,1]$ and $\pi_1(c,\cdot)$ is uniform over the actions for every context $c \in \mathcal{C}$, we can apply the fundamental inequality of online mirror descent for the KL divergence (Theorem 10.4 in (Orabona, 2019), V1), to obtain the following for each fixed context $c \in \mathcal{C}$ separately.

$$\sum_{k=1}^{K}\left\langle\hat{\ell}_k(c,\cdot),\pi_k(c,\cdot) - \pi_\star(c,\cdot)\right\rangle \leq \frac{\log|\mathcal{A}|}{\eta} + \frac{\eta}{2}\sum_{k=1}^{K}\pi_k(c,a)\underbrace{\hat{\ell}_k(c,a)^2}_{\leq 1} \leq \frac{\log|\mathcal{A}|}{\eta} + \frac{\eta K}{2}.$$

For $\eta = \sqrt{2\log|\mathcal{A}|/K}$ we obtain for each context $c \in \mathcal{C}$ that

$$\sum_{k=1}^{K}\left\langle\hat{\ell}_k(c,\cdot),\pi_k(c,\cdot) - \pi_\star(c,\cdot)\right\rangle \leq O(\sqrt{K\log|\mathcal{A}|}).$$

Since inner product of real vectors is symmetric,

$$\sum_{k=1}^{K} \left\langle \pi_k(c, \cdot) - \pi_\star(c, \cdot), \hat{\ell}_k(c, \cdot) \right\rangle \leq O(\sqrt{K \log |\mathcal{A}|}).$$

By taking an expectation over the contexts on both sides of the inequality, using linearity of expectation we obtain

$$\sum_{k=1}^{K} \mathbb{E}_{c_k} \left[ \left\langle \pi_k(c_k, \cdot) - \pi_\star(c_k, \cdot), \hat{\ell}_k(c_k, \cdot) \right\rangle \right] \leq O(\sqrt{K \log |\mathcal{A}|}),$$

as desired. $\qquad \square$

Lastly, we bound term (4).

**Lemma A.4** (Term (4) bound, restatment of Lemma 4.5)**.** *For any $\delta \in (0, 1)$ let $\beta = \sqrt{\frac{34K \log(4|\mathcal{F}|K^3/\delta)}{|\mathcal{A}|}}$. Then, the following holds with probability at least $1 - \delta/4$.*

$$\sum_{k=1}^{K} \mathbb{E}_{c_k} \left[ \left\langle \pi_\star(c_k, \cdot), \hat{\ell}_k(c_k, \cdot) - \ell(c_k, \cdot) \right\rangle \right] \leq \widetilde{O}\left( \sqrt{K|\mathcal{A}| \log(|\mathcal{F}|/\delta)} \right).$$

*Proof.* The following holds with probability at least $1 - \delta/4$.

$$\sum_{k=1}^{K} \mathbb{E}_{c_k} \left[ \left\langle \pi_\star(c_k, \cdot), \hat{\ell}_k(c_k, \cdot) - \ell(c_k, \cdot) \right\rangle \right]$$

$$\leq \sum_{k=2}^{K} \mathbb{E}_{c_k} \left[ \left\langle \pi_\star(c_k, \cdot), \hat{\ell}_k(c_k, \cdot) - \ell(c_k, \cdot) \right\rangle \right] + 1$$

$$= \sum_{k=2}^{K} \mathbb{E}_{c_k} \left[ \sum_{a \in A} \pi_\star(c_k, a) \left( \max\{0, \hat{f}_k(c_k, a) - b_k^\beta(c_k, a)\} - \ell(c_k, a) \right) \right] + 1$$

$$= \sum_{k=2}^{K} \mathbb{E}_{c_k} \left[ \sum_{a \in A} \pi_\star(c_k, a) \max\{0 - \ell(c_k, a), \hat{f}_k(c_k, a) - \ell(c_k, a) - b_k^\beta(c_k, a)\} \right] + 1$$

$$= \sum_{k=2}^{K} \mathbb{E}_{c_k} \left[ \sum_{a \in A} \pi_\star(c_k, a) \max\left\{ \underbrace{-f_\star(c_k, a)}_{=: \alpha_{k,a}}, \underbrace{\hat{f}_k(c_k, a) - f_\star(c_k, a) - b_k^\beta(c_k, a)}_{=: \rho_{k,a}} \right\} \right] + 1 = (\star).$$

We now note that for all $k \geq 2, a \in \mathcal{A}$ it holds that $\alpha_{k,a} \leq 0$, as $f_\star \in [0, 1]$. We next upper bound $\rho_{k,a}$ and then use the following fact to obtain the final upper bound.

*Fact* A.5. For any $x, y, z \in \mathbb{R}$ such that $z \geq y$ it holds that $\max\{x, y\} \leq \max\{x, z\}$.

Hence, we continue by upper bounding $\rho_{k,a}$, for any fixed $k \geq 2$ and $a \in \mathcal{A}$.

$$\rho_{k,a} = \left( \hat{f}_k(c_k, a) - f_\star(c_k, a) \right) - b_k^\beta(c_k, a)$$

$$\leq |\hat{f}_k(c_k, a) - f_\star(c_k, a)| - b_k^\beta(c_k, a)$$

$$= \min\left\{ 1, |\hat{f}_k(c_k, a) - f_\star(c_k, a)| \right\} - b_k^\beta(c_k, a) \quad \text{(Since the functions are bounded in } [0, 1] \text{ so is the absolute value)}$$

$$= \min\left\{ 1, \sqrt{\frac{\beta}{\beta} \frac{1 + \sum_{i=1}^{k-1} \pi_i(c_k, a)}{1 + \sum_{i=1}^{k-1} \pi_i(c_k, a)}} |\hat{f}_k(c_k, a) - f_\star(c_k, a)| \right\} - b_k^\beta(c_k, a)$$

$$\leq \min\left\{1, \frac{\beta/2}{1+\sum_{i=1}^{k-1}\pi_i(c_k,a)} + \frac{1}{2\beta}\left(1+\sum_{i=1}^{k-1}\pi_i(c_k,a)\right)\left(\hat{f}_k(c_k,a) - f_\star(c_k,a)\right)^2\right\} - b_k^\beta(c_k,a)$$

$$\text{(By AM-GM inequality)}$$

$$\leq \underbrace{\min\left\{1, \frac{\beta/2}{1+\sum_{i=1}^{k-1}\pi_i(c_k,a)}\right\}}_{=b_k^\beta(c_k,a)} + \frac{1}{2\beta}\left(1+\sum_{i=1}^{k-1}\pi_i(c_k,a)\right)\left(\hat{f}_k(c_k,a) - f_\star(c_k,a)\right)^2 - b_k^\beta(c_k,a)$$

$$\text{(Since all terms in the } \min \text{ are positive for } \beta > 0, \text{ holds by } \min\{\cdot,\cdot\} \text{ properties)}$$

$$= \frac{1}{2\beta}\left(1+\sum_{i=1}^{k-1}\pi_i(c_k,a)\right)\left(\hat{f}_k(c_k,a) - f_\star(c_k,a)\right)^2.$$

Hence, we choose $\xi_{k,a} := \frac{1}{2\beta}\left(1+\sum_{i=1}^{k-1}\pi_i(c_k,a)\right)\left(\hat{f}_k(c_k,a) - f_\star(c_k,a)\right)^2$, and note that for $\beta > 0$ we have $\xi_{k,a} \geq 0$. We then apply Fact A.5 to upper bound $(\star)$. We obtain

$$(\star) = \sum_{k=2}^{K}\mathbb{E}_{c_k}\left[\sum_{a\in A}\pi_\star(c_k,a)\max\left\{\underbrace{-f_\star(c_k,a)}_{\alpha_{k,a}}, \underbrace{\hat{f}_k(c_k,a) - f_\star(c_k,a) - b_k^\beta(c_k,a)}_{\rho_{k,a}}\right\}\right] + 1$$

$$\leq \sum_{k=2}^{K}\mathbb{E}_{c_k}\left[\sum_{a\in A}\pi_\star(c_k,a)\max\left\{\underbrace{-f_\star(c_k,a)}_{\leq 0}, \underbrace{\frac{1}{2\beta}\left(1+\sum_{i=1}^{k-1}\pi_i(c_k,a)\right)\left(\hat{f}_k(c_k,a) - f_\star(c_k,a)\right)^2}_{\xi_{k,a}\geq 0}\right\}\right] + 1$$

$$\text{(Since } \xi_{k,a} \geq \rho_{k,a})$$

$$= \sum_{k=2}^{K}\mathbb{E}_{c_k}\left[\sum_{a\in A}\pi_\star(c_k,a)\frac{1}{2\beta}\left(1+\sum_{i=1}^{k-1}\pi_i(c_k,a)\right)\left(\hat{f}_k(c_k,a) - f_\star(c_k,a)\right)^2\right] + 1$$

$$\leq \frac{K}{2\beta} + \frac{1}{2\beta}\sum_{k=2}^{K}\mathbb{E}_{c_k}\left[\sum_{a\in A}\underbrace{\pi_\star(c_k,a)}_{\leq 1}\sum_{i=1}^{k-1}\pi_i(c_k,a)\left(\hat{f}_k(c_k,a) - f_\star(c_k,a)\right)^2\right] + 1$$

$$\leq \frac{K}{2\beta} + \frac{1}{2\beta}\sum_{k=2}^{K}\mathbb{E}_{c_k}\left[\sum_{i=1}^{k-1}\sum_{a\in A}\pi_i(c_k,a)\left(\hat{f}_k(c_k,a) - f_\star(c_k,a)\right)^2\right] + 1$$

$$= \frac{K}{2\beta} + \frac{1}{2\beta}\sum_{k=2}^{K}\sum_{i=1}^{k-1}\mathbb{E}_{c_i}\left[\mathbb{E}_{a\sim\pi_i(c_i,\cdot)}\left(\hat{f}_k(c_i,a) - f_\star(c_i,a)\right)^2\right] + 1$$

$$\text{(By linearity of expectation, as the contexts are iid)}$$

$$\leq \frac{K}{2\beta} + \frac{68\log(4|\mathcal{F}|K^3/\delta)K}{2\beta} + 1. \qquad \text{(Holds with probability at least } 1 - \delta/4 \text{ by Corollary 4.1)}$$

Finally, by setting $\beta = \sqrt{\frac{34K\log(4|\mathcal{F}|K^3/\delta)}{|\mathcal{A}|}}$ we obtain that term (4) is bounded as $\widetilde{O}\left(\sqrt{K|\mathcal{A}|\log(|\mathcal{F}|/\delta)}\right)$. $\qquad\qquad \square$

## B. Auxiliary Lemmas

### B.1. Online Mirror Descent

In the Online Mirror Descent (OMD) algorithm (Orabona, 2019), at each round the agent updates its decision by solving the regularized constrained optimization problem given in Equation (8).

$$x_{k+1} \in \underset{x\in\Delta(d)}{\arg\min}\, \eta\langle g_k, x - x_k\rangle + B_\psi(x, x_k), \tag{8}$$

where $x_{k+1}$ is the next decision point computed relative to the previous point $x_k$, and $B_\psi$ denotes the Bregman divergence with respect to a (strictly convex) function $\psi$. The vector $g_k$ is a (possibly estimated) subgradient of the loss function at time $k$.

In bandit algorithms such as `Hedge` and `EXP3`, this decision point corresponds to a probability distribution over arms from which the learner samples an action. Since the expected loss is a linear function of the selected action distribution, $g_k$ is the estimated loss vector.

In our case, in addition to the above, $\psi$ is the negative entropy so that the associated Bregman divergence term $B_\psi$ is the KL divergence. Consequently, the `OMD` update reduces to the optimization problem given in Equation (7).

The following lemma states the fundamental inequality of `OMD` over the simplex with the KL divergence which will be used for our analysis.

**Theorem B.1** (Lemma 16 in (Shani et al., 2020), originally Theorem 10.4 in V1 of (Orabona, 2019), Fundamental inequality of Online Mirror Descent). *Assume for $g_{k,i} \geq 0$ for $k = 1, \ldots, K$ and $i = 1, \ldots, d$. Let $C = \Delta_d$ and $\eta > 0$. Using OMD with the KL-divergence, learning rate $\eta$, and with uniform initialization $x_1 = (1/d, \ldots, 1/d)$, the following holds for any $u \in \Delta_d$,*

$$\sum_{k=1}^{K} \langle g_k, x_k - u \rangle \leq \frac{\log d}{\eta} + \frac{\eta}{2} \sum_{k=1}^{K} \sum_{i=1}^{d} x_{k,i} g_{k,i}^2.$$

### B.2. Concentration Inequalities

**Theorem B.2** (Azuma-Hoeffding's inequality). *Let $(X_i)_{i=1}^{N}$ be a martingale difference sequence with respect to the filtration $(\mathcal{F}_i)_{i=0}^{N}$ such that $|X_i| \leq B$ almost surely for all $i \in [N]$. Then with probability at least $1 - \delta$,*

$$\left| \sum_{i=1}^{N} X_i \right| \leq B \sqrt{2N \log \frac{2}{\delta}}.$$

**Corollary B.3** (High probability regret). *With probability at least $1 - \delta/2$, it holds that*

$$\mathcal{R}_K \leq \mathbb{E}\mathcal{R}_K + 2\sqrt{2K \log(4/\delta)}.$$

*Proof.* The proof follows directly from Theorem B.2, as the contexts are stochastic and sampled iid in each round, and the policies $\{\pi_k\}_{k=2}^{K}$ are determined completely by the history, where $\pi_1$ is set to be the random policy. Thus, we have that

$$X_k := \langle \pi_k(c_k, \cdot) - \pi_\star(c_k, \cdot), \ell(c_k, \cdot) \rangle - \mathbb{E}_{c_k}[\langle \pi_k(c_k, \cdot) - \pi_\star(c_k, \cdot), \ell(c_k, \cdot) \rangle]$$

defines a martingale difference sequence $(X_k)_{k=1}^{K}$ with respect to the filtration $H_k = \{(c_i, a_i, \ell_i)\}_{i=1}^{k}$ which is the history up to (including) time $k$, for all $k \in [1, K]$ and $H_0 = \emptyset$ is the empty history. This holds true since $X_k$ is determined by $H_k$ for all $k$, and

$$\begin{aligned}
\mathbb{E}[X_k | H_{k-1}] =& \mathbb{E}[\langle \pi_k(c_k, \cdot) - \pi_\star(c_k, \cdot), \ell(c_k, \cdot) \rangle - \mathbb{E}_{c_k}[\langle \pi_k(c_k, \cdot) - \pi_\star(c_k, \cdot), \ell(c_k, \cdot) \rangle] | H_{k-1}] \\
=& \mathbb{E}_{c_k}[\langle \pi_k(c_k, \cdot) - \pi_\star(c_k, \cdot), \ell(c_k, \cdot) \rangle | H_{k-1}] - \mathbb{E}_{c_k}[\langle \pi_k(c_k, \cdot) - \pi_\star(c_k, \cdot), \ell(c_k, \cdot) \rangle | H_{k-1}] \\
=& 0.
\end{aligned}$$

In addition, $|X_k| \leq 2$ for all $k$. Hence, we can apply Theorem B.2 and obtain for any fixed $K \in \mathbb{N}$, when summing over $k = 1, 2, \ldots, K$, that the following holds with probability at least $1 - \delta/2$,

$$|\sum_{k=1}^{K} X_k| \leq 2\sqrt{2K \log(4/\delta)}.$$

In addition,

$$|\sum_{k=1}^{K} X_k| = |\sum_{k=1}^{K} (\langle \pi_k(c_k, \cdot) - \pi_\star(c_k, \cdot), \ell(c_k, \cdot) \rangle - \mathbb{E}_{c_k}[\langle \pi_k(c_k, \cdot) - \pi_\star(c_k, \cdot), \ell(c_k, \cdot) \rangle])|$$

$$=|\sum_{k=1}^{K}\langle \pi_k(c_k,\cdot) - \pi_\star(c_k,\cdot), \ell(c_k,\cdot)\rangle - \sum_{k=1}^{K}\mathbb{E}_{c_k}[\langle \pi_k(c_k,\cdot) - \pi_\star(c_k,\cdot), \ell(c_k,\cdot)\rangle]|$$

$$=|\mathcal{R}_K - \mathbb{E}\mathcal{R}_K|$$

Hence, with probability at least $1 - \delta/2$,

$$|\mathcal{R}_K - \mathbb{E}\mathcal{R}_K| \le 2\sqrt{2K\log(4/\delta)},$$

which implies the corollary. □

## B.3. Oracle Convergence

Lemma 5 in Xu and Zeevi (2020) presents a uniform convergence guarantee for offline lease-square regression.

**Lemma B.4** (uniform convergence over all sequences of estimators, Lemma 5 in (Xu and Zeevi, 2020)). *For an arbitrary contextual bandit algorithm, for all $\delta \in (0,1)$, with probability at least $1 - \delta/2$,*

$$\sum_{i=1}^{t-1}\mathbb{E}_{c_i,a_i}\left[(f_t(c_i,a_i) - f_\star(c_i,a_i))^2 \mid H_{i-1}\right]$$

$$\le 68\log(2|\mathcal{F}|t^3/\delta) + 2\sum_{i=1}^{t-1}(f_t(c_i,a_i) - \ell_i)^2 - (f_\star(c_i,a_i) - \ell_i)^2,$$

*uniformly over all $t \ge 2$ and all fixed sequence $f_2, f_3, \ldots \in \mathcal{F}$.*

In this lemma, the filtration defined by $H_i := \{(c_k,a_k,\ell_k)\}_{k=1}^{i}$, which is the history of observations up to time $i$.

The following is a direct corollary of Lemma B.4, which applies to the sequence of least-squares minimizers.

**Corollary B.5** (uniform convergence of offline least-squares regression, restatement of Corollary 4.1). *Let $f_2, f_3, \ldots \in \mathcal{F}$ denote the sequence of least squares minimizers and let $\pi_1, \pi_2, \ldots$ denote the sequence of played contextual policies. The following holds for any $\delta \in (0,1)$ and $t \ge 2$ with probability at least $1 - \delta/4$.*

$$\sum_{i=1}^{t-1}\mathbb{E}_{c_i}\left[\mathbb{E}_{a_i \sim \pi_i(c_i,\cdot)}\left[(f_t(c_i,a_i) - f_\star(c_i,a_i))^2\right]\right] \le 68\log(4|\mathcal{F}|t^3/\delta).$$

*Proof.* For the sequence of least squares minimizers, for all $t \ge 2$, it holds that

$$\sum_{i=1}^{t-1}(f_t(c_i,a_i) - \ell_i)^2 - (f_\star(c_i,a_i) - \ell_i)^2 \le 0.$$

In addition, in each round $k$, given the history $H_{k-1} = \{(c_i,a_i,\ell_i)\}_{i=1}^{k-1}$, we have that $\pi_k$ is determined completely. Hence, the corollary follows from Lemma B.4 for the choice in $\delta' = \delta/2$. □

## B.4. Additional Algebraic Lemmas

**Lemma B.6** (Lemma C.1 in Levy et al. (2024)). *Let $S_t = \lambda + \sum_{k=1}^{t-1} x_k$ and $x_t \in [0,\lambda]$ for all $t$. Then*

$$\sum_{t=1}^{T}\frac{x_t}{S_t} \le 2\log(T+1).$$

# C. Experiments

As is standard in the CMAB literature, we evaluate the performance of our algorithm using the Vowpal Wabbit[5] benchmark suite (Bietti et al., 2021), which implements all known feasibly-implementable CMAB algorithms or their practical variants.

Due to the diversity of algorithm implementations in the library, some methods require a cost-sensitive classification oracle (e.g., `OnlineCover`-the heuristics of `ILTCB` (Agarwal et al., 2014)), while others rely on an online regression oracle (`SquareCB` (Foster and Rakhlin, 2020), `AdaCB` (Foster et al., 2021b), `RegCB` (Foster et al., 2018a) and `FastCB` (Foster and Krishnamurthy, 2021)). We emphasize that even algorithms originally designed for offline regression oracles can be effectively executed using an online regression oracle, which is a stronger oracle capable of handling adversarial examples (Cesa-Bianchi and Lugosi, 2006; Foster and Rakhlin, 2020). In the Vowpal Wabbit implementation, all algorithms that are using a regression oracle are implemented using its online variant, which is the base learner of this library. Following this convention, we implement `OPO-CMAB` using the base online regression oracle provided by the library.

The state-of-the-art empirical evaluation of CMAB algorithms on the Vowpal Wabbit benchmark is presented in Foster and Krishnamurthy (2021). They conduct extensive experiments comparing all known efficient CMAB algorithms based on regression oracles (e.g., `SquareCB`, `AdaCB`, `RegCB`) as well as a supervised learning algorithm as a mild baseline, against their proposed algorithm, `FastCB`. The evaluation spans a wide range of hyperparameter settings and considers both square loss and logistic loss for the regression oracle. Their findings indicate that `FastCB` with logistic loss achieves the best performance, followed by `SquareCB` with logistic and squared loss in second and third place, respectively. `AdaCB` and `RegCB` perform significantly worse. Based on these results, we adopt the same experimental setup and focus our comparison on the regression-based candidates: `FastCB`, `SquareCB`, `RegCB` and `AdaCB` as well as `Supervised` which is the supervised learning baseline. More details about the used implementations and hyperparameter are given in the sequel.

## C.1. Function classes and Oracles

For the function class and oracle, we follow the same setup as in Foster and Krishnamurthy (2021) that is also described in detail in Bietti et al. (2021). We give the details below for completeness.

**Oracle.** All evaluated algorithms, including our `OPO-CMAB`, are implemented in Vowpal Wabbit (VW), using its base online learning procedure, regardless of whether they are designed for online or offline regression oracles. This procedure implements Importance Weighted Regression (IWR) for both square loss and logistic loss, as described in Bietti et al. (2021), i.e., the oracle performs Online Gradient Descent (OGD) (Bartlett et al., 2007) updates with an adaptive step size that is controlled by the learning-rate hyperparameter, following Duchi et al. (2011), normalization as in Ross et al. (2013), and importance weighting as in Karampatziakis and Langford (2011). In addition, VW applies a Cost-Sensitive One-Against-All (CSOAA) reduction for multiclass classification, which ultimately yields label predictions for each action. All tested algorithms are constrained to use this IWR-based oracle exclusively, which in the code reflected by the 'mtr' cb-type parameter.

**Losses and Function Class.** Regarding the losses and function class used, we again adopt the same setup tested by Foster and Krishnamurthy (2021); Bietti et al. (2021). Hence, we test all the algorithms when applying the online logistic regression oracle using the log loss. We also test all of the algorithms, excluding `FastCB`, using the square loss. The two mathematically defined bellow, for a true label $y$ and predicted label $\hat{y}$.

$$\ell_{\text{sq}}(\hat{y}, y) = (\hat{y} - y)^2.$$
$$\ell_{\text{log}}(\hat{y}, y) = y \log(1/\hat{y}) + (1 - y) \log(1/1 - \hat{y}).$$

We note that the log-loss (or cross-entropy) in this notation is defined for binary classification, i.e., $y, \hat{y} \in \{0, 1\}$, and we next extend it to multiclass classification using the sigmoid link-function, which formally defined as

$$\sigma(x) = \frac{1}{1 + \exp(-x)}.$$

Then, $\ell_{\text{logistic}}(\hat{y}, y) = \ell_{\text{log}}(\sigma(\hat{y}), y)$. See Foster et al. (2018b) for more information regarding this equivalence. When applying logistic regression, we define the function class $\mathcal{F}$ as the class of generalized linear models, formally given by $\mathcal{F} = \{(c, a) \to \sigma(\langle w, \phi(c, a) \rangle) | w \in \mathbb{R}^d\}$, where $\sigma(\cdot)$ is the logistic link function defined above, and $\phi : \mathcal{C} \times \mathcal{A} \to \mathbb{R}^d$ is a dataset-related feature map.

---

[5] https://vowpalwabbit.org/

When using the squared loss, we instead choose $\mathcal{F}$ to be the class of linear functions

$$\mathcal{F} = \left\{ (c, a) \to \langle w, \phi(c, a) \rangle | w \in \mathbb{R}^d \right\}.$$

All of these implementation choices are already supported by the Vowpal Wabbit library, and also performed by Foster and Krishnamurthy (2021)[6].

### C.2. Implementation Details

Bellow we elaborate on the implementation and hyper-parameters tested for each algorithm.

**Implementation of known algorithms.** For the implementations and experimental setup of `SquareCB`, `FastCB`, `Supervised` we use the source code provided by Foster and Krishnamurthy (2021)[7]. We use the hyperparameter values tested in this work, among them $\rho \in \{0.25, 0.5\}, \gamma_0 \in \{1000, 700, 400, 100, 50, 10\}$. Due to limited computational resources, we use as learning rates a subset of the learning rates tested by Foster and Krishnamurthy (2021) and mentioned in Table 1.
For `RegCB` we use the standard implementation of it in Vowpal Wabbit, with the hyper-parameters choice of $c_0 \in 10^{\{-1, -2, -3\}}$ suggested by Bietti et al. (2021).
For `AdaCB`, we use the `SquareCB` implementation of Vowpal Wabbit, which includes elimination option that uses confidence bounds computed as for `RegCB` to eliminate sub-optimal actions, and then applies Inverse Gap Weightening (IGW)-based policy for the action selection itself. We test the algorithm on the set of hyperparameter suggested by Foster et al. (2021b): $c_0 \in 10^{\{-1, -2, -3\}}, \rho \in \{0.25, 0.5\}, \gamma_0 \in \{1000, 700, 400, 100, 50, 10\}$. For a summary of all hyper-parameter values used for tuning, see Table 1.

As for the implementation of `OPO-CMAB`, we integrate it into the same source code ourself, and the implementation details are given next.

**`OPO-CMAB` implementation.** We integrate `OPO-CMAB` into the Vowpal Wabbit library, implementing the exact pseudo-code provided in Algorithm 1, with the following practical modifications:
We define the bonus factor at each round $k$ as $\beta_k = \gamma \sqrt{\frac{k}{|\mathcal{A}|}}$, where $\gamma$ is a tuned hyperparameter. This differs from the constant $\beta = \sqrt{\frac{34K \log(4|\mathcal{F}|K^3/\delta)}{|\mathcal{A}|}}$ used in our theoretical analysis. This change is motivated by the following:

1. *Adaptivity to unknown horizon.* The new form of $\beta_k$ allows the algorithm to operate without prior knowledge of the total number of rounds $K$.

2. *Practical tuning and computational efficiency.* The constant term in the original $\beta$ expression, $\sqrt{34 \log(4|\mathcal{F}|K^3/\delta)}$, may be difficult to compute when $\mathcal{F}$ is infinite or extremely large. We replace it with a tunable parameter $\gamma$ to allow for more practical and flexible implementation of the confidence bounds width factor.

Similarly, we treat the parameter $\eta$ as a tuned hyperparameter rather than constant that depends on $K, |\mathcal{A}|$, for the same reasons as above. We note that our algorithm, similarly to the others, have an additional learning rate parameter, that uses the base-learner oracle as an initial step size. We, similarly to the other algorithm, refer the oracle's learning rate as a tuned hyper-parameter.

An important difference between our algorithm and the others is that `OPO-CMAB` requires evaluating each new context using all past predictors, in order to compute exploration bonuses based on past counterfactual policies. To enable this functionality without interfering with the internal implementation of the base learner, we cache the learned weights at each round and use them to generate past predictions for the current context. This memory and run time-consuming implementation is due to the constraints implied by the oracle implementation of Vowpal Wabbit library.

Lastly, we remark that although `OPO-CMAB` was originally designed for the squared loss, the dataset used in this experimental setup involves multi-class and multi-label classification. In such settings, the logistic loss is often more appropriate, as it better reflects the probabilistic nature of the labels and can yield more accurate loss estimates in practice compared to the squared loss. Hence, we apply our algorithm using the logistic-loss as well, to provide it with more accurate predictors.

---

[6]For additional details, see Appendix D.2 in Foster and Krishnamurthy (2021).
[7]The code is publicly available at https://openreview.net/forum?id=3qYgdGj9Svt

Apart from these changes, the rest of the implementation adheres closely to the description in Algorithm 1. The hyper-parameter values used for tuning `OPO-CMAB` also appear in Table 1.

*Remark* C.1. We note that the adaptive $\beta_k = O\left(\gamma\sqrt{\frac{k}{|\mathcal{A}|}}\right)$ choice can be proven to yield optimal regret bound for stochastic CMAB (see Xu and Zeevi (2020) for more information), however, for clean representation of the algorithm and main theoretical result, we chose in a static $\beta$ parameter.

### C.3. Experiments Description and Evaluation

In the following, we describe our experimental setup. Due to computational limitations, we restrict our evaluation to the following experimental setup.

**Tested Datasets.** We evaluated all the considered algorithms on 18 relatively-small size selected out of the large set of 515 multiclass classification datasets defined in the Vowpal Wabbit suite[8]. The tested datasets are specified in Table 2. In each dataset, every context is associated with a true label. Following prior works, we simulate bandit feedback by defining the agent's loss to be 0 if it predicts the correct label, and 1 otherwise.

**Performance Evaluation.** We again follow Foster and Krishnamurthy (2021); Bietti et al. (2021) and evaluate the performance of each algorithm on a given dataset using the final *Progressive Validation (PV)* loss (Blum et al., 1999), which, for an algorithm A and a given dataset containing $K$ examples, formally defined as

$$L_{PV}(\mathtt{A}, K) = \frac{1}{K}\sum_{k=1}^{K}\ell_k(a_k).$$

In our experiments, we measure the decrease in the PV-loss as a function of the number of examples.

*Remark* C.2. For any two algorithms A, B, let $\mathcal{R}_K^{\mathtt{A}}$ and $\mathcal{R}_K^{\mathtt{B}}$ denote their regret computed using the binary loss defined above on a given dataset. Then, it holds true that

$$\frac{1}{K}\left(\mathcal{R}_K^{\mathtt{A}} - \mathcal{R}_K^{\mathtt{B}}\right) = L_{PV}(\mathtt{A}, K) - L_{PV}(\mathtt{B}, K).$$

Hence, in this setting, the PV-loss is a representative quantity for the regret differences between any pair of algorithms.

In addition, we also consider the difference of the final averaged-across-permutations PV-loss of each algorithm compare to that of the supervised learning mild baseline.

**Hyper-parameters Tuning.** We tuned all algorithms by running all the combinations of the the parameters specified in Table 1. For each algorithm, we run each combination once on each dataset, and choose the combination that provides the lowest final PV-loss. Then, we use only this best combination to run our experiments. The chosen parameter configurations are specified in Table 1.

| Algorithm | $\gamma/\gamma_0$ | $\rho$ | $c_0$ | $\eta$ | Loss | LRs |
|---|---|---|---|---|---|---|
| Supervised | – | – | – | – | $\ell_{\text{logistic}}$ | $10^{\{1,0,-1,-2,-3\}}$ |
| SquareCB | 1000, 700, 400, 100, 50, 10 | 0.5, 0.25 | – | – | $\ell_{\text{sq}}, \ell_{\text{logistic}}$ | $10^{\{1,0,-1,-2,-3\}}$ |
| FastCB | 1000, 700, 400, 100, 50, 10 | 0.5, 0.25 | – | – | $\ell_{\text{logistic}}$ | $10^{\{1,0,-1,-2,-3\}}$ |
| AdaCB | 1000, 700, 400, 100, 50, 10 | 0.5, 0.25 | $10^{\{-1,-2,-3\}}$ | – | $\ell_{\text{sq}}, \ell_{\text{logistic}}$ | $10^{\{1,0,-1,-2,-3\}}$ |
| RegCB | – | – | $10^{\{-1,-2,-3\}}$ | – | $\ell_{\text{sq}}, \ell_{\text{logistic}}$ | $10^{\{1,0,-1,-2,-3\}}$ |
| OPO-CMAB | $10^{\{0,-1,-2\}}$ | – | – | 100, 10, 1, 0.2, 0.1, 0.01 | $\ell_{\text{sq}}, \ell_{\text{logistic}}$ | $10^{\{1,0,-1,-2,-3\}}$ |

*Table 1.* All tested hyperparameter values per algorithm.

### C.4. Experiments and Results

**Experimental setup and results.** For each algorithm and dataset, we run the best hyperparameter configuration (as found in our tuning) on 10 random permutations of the dataset. The results are summarized in the following tables and plots.

First, we average the PV-loss at each time step across permutations to measure the average performance of each algorithm as a function of the number of observed examples. In Figure 1, we show the averaged PV-loss curves as well as the Std on three

---

[8]All datasets are publicly available to download from OpenML collection `https://www.openml.org`

| Dataset | Supervised | SquareCB | FastCB | AdaCB | RegCB | OPOCMAB |
|---|---|---|---|---|---|---|
| 1006 | $\ell_{\text{logistic}}$, lr: 10 | $\rho$: 0.5, $\gamma_0$: 10, $\ell_{\text{logistic}}$, lr: 10 | $\rho$: 0.25, $\gamma_0$: 50, $\ell_{\text{logistic}}$, lr: 10 | $\rho$: 0.5, $\gamma_0$: 100, $c_0$: 0.001, $\ell_{\text{logistic}}$, lr: 0.01 | $c_0$: 0.001, $\ell_{\text{sq}}$, lr: 1 | $\eta$: 100, $\gamma$: 1, $\ell_{\text{logistic}}$, lr: 0.1 |
| 1012 | $\ell_{\text{logistic}}$, lr: 1 | $\rho$: 0.5, $\gamma_0$: 50, $\ell_{\text{logistic}}$, lr: 0.1 | $\rho$: 0.5, $\gamma_0$: 1000, $\ell_{\text{logistic}}$, lr: 0.001 | $\rho$: 0.5, $\gamma_0$: 50, $c_0$: 0.01, $\ell_{\text{logistic}}$, lr: 0.1 | $c_0$: 0.1, $\ell_{\text{logistic}}$, lr: 0.1 | $\eta$: 1, $\gamma$: 0.01, $\ell_{\text{sq}}$, lr: 0.1 |
| 1015 | $\ell_{\text{logistic}}$, lr: 0.1 | $\rho$: 0.25, $\gamma_0$: 1000, $\ell_{\text{sq}}$, lr: 10 | $\rho$: 0.25, $\gamma_0$: 100, $\ell_{\text{logistic}}$, lr: 0.1 | $\rho$: 0.5, $\gamma_0$: 100, $c_0$: 0.1, $\ell_{\text{logistic}}$, lr: 0.1 | $c_0$: 0.001, $\ell_{\text{sq}}$, lr: 1 | $\eta$: 100, $\gamma$: 0.1, $\ell_{\text{sq}}$, lr: 0.1 |
| 1062 | $\ell_{\text{logistic}}$, lr: 1 | $\rho$: 0.25, $\gamma_0$: 10, $\ell_{\text{logistic}}$, lr: 1 | $\rho$: 0.25, $\gamma_0$: 400, $\ell_{\text{logistic}}$, lr: 0.1 | $\rho$: 0.5, $\gamma_0$: 100, $c_0$: 0.01, $\ell_{\text{logistic}}$, lr: 0.1 | $c_0$: 0.001, $\ell_{\text{logistic}}$, lr: 0.01 | $\eta$: 100, $\gamma$: 0.01, $\ell_{\text{sq}}$, lr: 1 |
| 1073 | $\ell_{\text{logistic}}$, lr: 0.01 | $\rho$: 0.25, $\gamma_0$: 10, $\ell_{\text{logistic}}$, lr: 1 | $\rho$: 0.5, $\gamma_0$: 10, $\ell_{\text{logistic}}$, lr: 10 | $\rho$: 0.25, $\gamma_0$: 1000, $c_0$: 0.1, $\ell_{\text{logistic}}$, lr: 0.001 | $c_0$: 0.001, $\ell_{\text{sq}}$, lr: 10 | $\eta$: 1, $\gamma$: 0.01, $\ell_{\text{logistic}}$, lr: 1 |
| 1084 | $\ell_{\text{logistic}}$, lr: 0.001 | $\rho$: 0.5, $\gamma_0$: 100, $\ell_{\text{sq}}$, lr: 0.1 | $\rho$: 0.5, $\gamma_0$: 10, $\ell_{\text{logistic}}$, lr: 0.001 | $\rho$: 0.5, $\gamma_0$: 50, $c_0$: 0.1, $\ell_{\text{logistic}}$, lr: 0.1 | $c_0$: 0.1, $\ell_{\text{logistic}}$, lr: 0.001 | $\eta$: 10, $\gamma$: 0.01, $\ell_{\text{logistic}}$, lr: 0.001 |
| 339 | $\ell_{\text{logistic}}$, lr: 1 | $\rho$: 0.5, $\gamma_0$: 700, $\ell_{\text{sq}}$, lr: 0.001 | $\rho$: 0.5, $\gamma_0$: 700, $\ell_{\text{logistic}}$, lr: 10 | $\rho$: 0.5, $\gamma_0$: 100, $c_0$: 0.1, $\ell_{\text{logistic}}$, lr: 0.01 | $c_0$: 0.1, $\ell_{\text{logistic}}$, lr: 10 | $\eta$: 100, $\gamma$: 0.01, $\ell_{\text{sq}}$, lr: 10 |
| 346 | $\ell_{\text{logistic}}$, lr: 1 | $\rho$: 0.5, $\gamma_0$: 1000, $\ell_{\text{logistic}}$, lr: 0.001 | $\rho$: 0.5, $\gamma_0$: 1000, $\ell_{\text{logistic}}$, lr: 0.001 | $\rho$: 0.5, $\gamma_0$: 50, $c_0$: 0.1, $\ell_{\text{logistic}}$, lr: 0.001 | $c_0$: 0.01, $\ell_{\text{logistic}}$, lr: 0.001 | $\eta$: 10, $\gamma$: 1, $\ell_{\text{logistic}}$, lr: 10 |
| 457 | $\ell_{\text{logistic}}$, lr: 1 | $\rho$: 0.5, $\gamma_0$: 100, $\ell_{\text{logistic}}$, lr: 0.01 | $\rho$: 0.25, $\gamma_0$: 700, $\ell_{\text{logistic}}$, lr: 0.001 | $\rho$: 0.5, $\gamma_0$: 1000, $c_0$: 0.01, $\ell_{\text{logistic}}$, lr: 0.001 | $c_0$: 0.1, $\ell_{\text{sq}}$, lr: 10 | $\eta$: 100, $\gamma$: 0.1, $\ell_{\text{logistic}}$, lr: 10 |
| 476 | $\ell_{\text{logistic}}$, lr: 0.1 | $\rho$: 0.5, $\gamma_0$: 700, $\ell_{\text{sq}}$, lr: 0.1 | $\rho$: 0.25, $\gamma_0$: 100, $\ell_{\text{logistic}}$, lr: 0.1 | $\rho$: 0.25, $\gamma_0$: 10, $c_0$: 0.001, $\ell_{\text{logistic}}$, lr: 1 | $c_0$: 0.001, $\ell_{\text{sq}}$, lr: 0.1 | $\eta$: 100, $\gamma$: 0.01, $\ell_{\text{sq}}$, lr: 10 |
| 729 | $\ell_{\text{logistic}}$, lr: 0.001 | $\rho$: 0.5, $\gamma_0$: 1000, $\ell_{\text{logistic}}$, lr: 0.1 | $\rho$: 0.5, $\gamma_0$: 1000, $\ell_{\text{logistic}}$, lr: 0.01 | $\rho$: 0.25, $\gamma_0$: 10, $c_0$: 0.01, $\ell_{\text{logistic}}$, lr: 0.001 | $c_0$: 0.1, $\ell_{\text{sq}}$, lr: 0.001 | $\eta$: 0.1, $\gamma$: 0.1, $\ell_{\text{sq}}$, lr: 1 |
| 785 | $\ell_{\text{logistic}}$, lr: 0.1 | $\rho$: 0.5, $\gamma_0$: 10, $\ell_{\text{sq}}$, lr: 0.1 | $\rho$: 0.5, $\gamma_0$: 10, $\ell_{\text{logistic}}$, lr: 0.1 | $\rho$: 0.5, $\gamma_0$: 700, $c_0$: 0.001, $\ell_{\text{logistic}}$, lr: 0.001 | $c_0$: 0.01, $\ell_{\text{sq}}$, lr: 0.1 | $\eta$: 1, $\gamma$: 1, $\ell_{\text{logistic}}$, lr: 10 |
| 835 | $\ell_{\text{logistic}}$, lr: 1 | $\rho$: 0.5, $\gamma_0$: 400, $\ell_{\text{sq}}$, lr: 0.001 | $\rho$: 0.5, $\gamma_0$: 1000, $\ell_{\text{logistic}}$, lr: 0.01 | $\rho$: 0.5, $\gamma_0$: 1000, $c_0$: 0.001, $\ell_{\text{logistic}}$, lr: 1 | $c_0$: 0.01, $\ell_{\text{logistic}}$, lr: 1 | $\eta$: 100, $\gamma$: 1, $\ell_{\text{sq}}$, lr: 1 |
| 848 | $\ell_{\text{logistic}}$, lr: 0.001 | $\rho$: 0.5, $\gamma_0$: 700, $\ell_{\text{logistic}}$, lr: 0.1 | $\rho$: 0.5, $\gamma_0$: 700, $\ell_{\text{logistic}}$, lr: 0.1 | $\rho$: 0.5, $\gamma_0$: 700, $c_0$: 0.1, $\ell_{\text{logistic}}$, lr: 0.01 | $c_0$: 0.1, $\ell_{\text{logistic}}$, lr: 1 | $\eta$: 100, $\gamma$: 0.01, $\ell_{\text{sq}}$, lr: 0.01 |
| 874 | $\ell_{\text{logistic}}$, lr: 1 | $\rho$: 0.5, $\gamma_0$: 400, $\ell_{\text{logistic}}$, lr: 0.01 | $\rho$: 0.5, $\gamma_0$: 1000, $\ell_{\text{logistic}}$, lr: 0.001 | $\rho$: 0.5, $\gamma_0$: 10, $c_0$: 0.01, $\ell_{\text{logistic}}$, lr: 10 | $c_0$: 0.1, $\ell_{\text{sq}}$, lr: 1 | $\eta$: 1, $\gamma$: 0.1, $\ell_{\text{sq}}$, lr: 1 |
| 905 | $\ell_{\text{logistic}}$, lr: 10 | $\rho$: 0.5, $\gamma_0$: 700, $\ell_{\text{logistic}}$, lr: 0.1 | $\rho$: 0.5, $\gamma_0$: 700, $\ell_{\text{logistic}}$, lr: 0.1 | $\rho$: 0.25, $\gamma_0$: 700, $c_0$: 0.001, $\ell_{\text{logistic}}$, lr: 0.001 | $c_0$: 0.01, $\ell_{\text{sq}}$, lr: 10 | $\eta$: 100, $\gamma$: 1, $\ell_{\text{sq}}$, lr: 1 |
| 928 | $\ell_{\text{logistic}}$, lr: 1 | $\rho$: 0.25, $\gamma_0$: 50, $\ell_{\text{logistic}}$, lr: 0.01 | $\rho$: 0.25, $\gamma_0$: 700, $\ell_{\text{logistic}}$, lr: 0.001 | $\rho$: 0.25, $\gamma_0$: 10, $c_0$: 0.01, $\ell_{\text{logistic}}$, lr: 0.001 | $c_0$: 0.1, $\ell_{\text{logistic}}$, lr: 0.001 | $\eta$: 0.1, $\gamma$: 0.1, $\ell_{\text{logistic}}$, lr: 1 |
| 964 | $\ell_{\text{logistic}}$, lr: 0.001 | $\rho$: 0.25, $\gamma_0$: 100, $\ell_{\text{sq}}$, lr: 0.01 | $\rho$: 0.5, $\gamma_0$: 700, $\ell_{\text{logistic}}$, lr: 0.01 | $\rho$: 0.5, $\gamma_0$: 50, $c_0$: 0.01, $\ell_{\text{logistic}}$, lr: 0.001 | $c_0$: 0.001, $\ell_{\text{logistic}}$, lr: 0.01 | $\eta$: 0.1, $\gamma$: 0.1, $\ell_{\text{sq}}$, lr: 10 |

*Table 2.* Best hyperparameter configurations per algorithm and dataset.

representative datasets, chosen because in them the supervised baseline converges to a significantly better-than-random-policy final PV-loss.

Figure 2 presents a table summarizing, for each dataset, the difference in final averaged-across-permutations PV-loss between each algorithm and the supervised baseline. The algorithm with the best mean difference relative to the supervised baseline is highlighted. Negative numbers indicate the algorithm outperforms supervised; positive numbers show how close it comes to the supervised baseline.

**Discussion on Figure 2 results.** Figure 2 shows that across all datasets, `AdaCB`, `FastCB`, `OPO-CMAB` and `RegCB` are closest to (and sometimes outperforms) the supervised baseline in 4 out of 18 datasets each, followed by `SquareCB` lag behind, with only 2 datasets out of the 18, for which is the closed to supervised. As in most datasets the difference between the tested algorithms are small, there is no clear evidence that one algorithm significantly outperform the others.

Overall, Figure 2 supports the hypothesis that all algorithms perform comparably considering those datasets, with no significant evidence that any one algorithm consistently outperforms the others.

## C.5. Resources and Computation

**Assets.** We used the following assets to conduct our experiments.
The code for the Vowpal Wabbit library, which is publicly available at https://vowpalwabbit.org/, includes implementations of all tested CMAB algorithms, with the exception of `FastCB`.

**Mean Differences from Supervised**

| Dataset | AdaCB | FastCB | OPOCMAB | RegCB | SquareCB |
|---------|-------|--------|---------|-------|----------|
| **1006_2** | 0.167567 | 0.122973 | 0.154054 | **0.108784** | 0.127702 |
| **1012_2** | 0.043814 | 0.043814 | 0.045361 | 0.046907 | **0.038144** |
| **1015_2** | **-0.013889** | -0.004167 | 0.044444 | 0.001389 | 0.004166 |
| **1062_2.0** | 0.022222 | **0.002778** | 0.013889 | 0.033333 | 0.044444 |
| **1073_2.0** | 0.027007 | **-0.064598** | 0.009854 | -0.060949 | -0.040876 |
| **1084_3** | 0.215000 | 0.028637 | **0.001818** | 0.009546 | 0.145000 |
| **339_3** | **0.036111** | 0.052778 | 0.063889 | 0.108333 | 0.069444 |
| **346_2** | **0.002000** | 0.004000 | 0.002000 | 0.018000 | 0.020000 |
| **457_4** | 0.200000 | 0.140741 | **0.062963** | 0.111111 | 0.096296 |
| **476_2** | 0.038000 | 0.046000 | 0.008000 | 0.022000 | **-0.012000** |
| **729_2** | 0.113636 | **0.009091** | 0.022727 | 0.056818 | 0.018182 |
| **785_2** | 0.048889 | 0.026667 | **0.006666** | 0.033333 | 0.011111 |
| **835_2** | 0.218750 | 0.200000 | **0.189583** | 0.218750 | 0.235417 |
| **848_2** | 0.084211 | 0.094737 | 0.142105 | **0.036842** | 0.089474 |
| **874_2** | 0.112000 | **0.054000** | 0.108000 | 0.116000 | 0.094000 |
| **905_2** | 0.064102 | 0.066667 | 0.164103 | **0.033333** | 0.066667 |
| **928_2** | **0.156522** | 0.200000 | 0.197826 | 0.223913 | 0.215217 |
| **964_2** | 0.077778 | 0.105555 | 0.122222 | **0.044444** | 0.166667 |

☐ Winner (Best Performance w.r.t. Supervised)

*Figure 2.* Mean Difference from Supervised baseline.

The implementation of `FastCB`, as well as the full experimental setup and evaluation code, were obtained from the source code accompanying Foster and Krishnamurthy (2021), which is publicly available at `https://openreview.net/forum?id=3qYgdGj9Svt`.

Building on this foundation, we extended the codebase by adding our own implementations and modifications. As noted by the Foster and Krishnamurthy (2021), their code builds upon the CMAB evaluation framework developed in Bietti et al. (2021), which is also publicly available at `https://github.com/vowpalwabbit/vowpal_wabbit/`.

Following previous works, we use in our experiments 18 datasets from the 515 multiclass and multi-label classification datasets used for CMAB evaluation from the OpenML collection. The datasets are publicly available for download at `https://www.openml.org`.

See dataset 1015 here: `https://www.openml.org/search?type=data&sort=runs&id=1015&status=active`, dataset 1062 here: `https://www.openml.org/search?type=data&sort=runs&id=1062&status=active`, and dataset 1084 here: `https://www.openml.org/search?type=data&sort=runs&id=1084&status=active`.

**Computation resources.** Experiments were conducted on a Linux CPU server with approximately 250 cores, from which we used 40-100 cores only. The total compute time required to run all the presented experiments was approximately 48 hours. The memory required to run the tests was about 1.5 TB.

