# OpenReview forum: "Optimal Regret for Policy Optimization in Contextual Bandits"
_ICML.cc/2026/Conference — ICML 2026 regular_

### Official Review · Reviewer_TPPa · 2026-02-23

**Soundness:** 3
**Presentation:** 2
**Significance:** 2
**Originality:** 3
**Overall Recommendation:** 3
**Confidence:** 4

**Summary:**

The paper studies $\textbf{stochastic contextual muti-armed bandits with general offline function approximation}$ under a realizability assumption. The author propose an $\textbf{optimistic policy optimization}$ approach that uses regression oracle to estimate the expected loss for each $(c, a)$ pair and constructs an optimistic loss by extracting exploration bonus. The policy is then updated directly from these optimistic losses via an exponential weights update. The proposed algorithm is claimed to achieve a regret bound matching the optimal order, namely $O\bigl(\sqrt{K \arrowvert A \arrowvert \log\arrowvert \mathcal{F} \arrowvert} \bigr)$ for a finite realizable class $\mathcal{F}$, where $K$ is number of rounds and $\arrowvert A \arrowvert$ is the number of actions. The authors also includes exmpricial comparison with several contextual bandits baselines on the benchmark datasets.

**Compliance With Llm Reviewing Policy:**

Affirmed.

**Final Justification:**

Weak reject. The rebuttal addresses my concerns, especially regarding Lemma A.2. However, I still find the infinite function class extension too imprecise, and the rebuttal does not materially change my view on practical significance or empirical advantage. Consider all weakness and strength I give weak reject

**Key Questions For Authors:**

1. In the proof of Lemma A.2 in the page $14$ (around line $722$ to $725$). The equality in this step does not seems to follow directly from linearity of expectation and the iid assumption on contexts. In particular,  $\hat{f}_k$ is fitted using the history up to round $k-1$, so  $\hat{f}_k(c_k, a)$ is evaluated on a fresh context $c_k$ (not used in fitting), whereas after the index change the term $\hat{f}_k(c_i, a)$ (for $i < k$) is evaluated on the past context $c_i$, which is the part of the historical used to construct $\hat{f}_k$. Since this step is essential for applying Corollary 4.1 later, could the authors provide a more rigorous justification for this step?

2. The conclusion claims that the results can be immediately extended to infinite function classes. Could the authors provide a more precise statement of this extension: a. what complexity measure is intended to replace the finite class $\log\arrowvert \mathcal{F} \arrowvert$ term b. would the regret bound retain the same form or would additional terms appear.

**Limitations:**

The paper is theoretical and impact is well discussed

**Strengths And Weaknesses:**

Strengths:
1. The paper approaches the stochastic contextual bandits from a relatively new angle by policy optimization using optimistic loss estimates

2. The proposed algorithm is conceptually simple and intuitive.

3. The claimed regret bound matches the optimal order achieved by exiting method in the same finite class realizable setting

4. The empirical study compares against several contextual bandits baselines and is generally fair in its setup and reporting.

Weakness:
1. The main algorithm takes the learning rate $\eta$ and exploration parameter $\beta$ as inputs. While the appendix later provides explicit closed-form choices, the main algorithm should state or directly reference these values to improve the clarity and readability.

2. The author claim an immediate extension to infinite contexts and infinite function classes. Extension to infinite contexts spaces maybe plausible, but extension to infinite function classes is not obvious. In particular, key steps like Corollary $4$. appear to rely on the finite class argument and it is unclear what complexity measure would replace the $\log\arrowvert \mathcal{F} \arrowvert$ term and how the regret bound would change in the infinite classes case.

3. Beyond the policy optimization perspective, it is not clear what practical or theoretical advantage is over existing methods: a. the claimed regret bound is same to prior work; b. the empirical results do not show a clear advantage; c. The paper reports a runtime of the order of $K^2$ (where $K$ is number of rounds), which is relatively high.

5. There is a potential gap in the proof. There appears to be a nontrivial issue in the proof of Lemma A.2 in the page $14$ (around line $722$ to $725$): the equality does not seems to follow directly from linearity of expectation and the iid assumption on contexts. More in Key Questions For Authors.

---

> ### Author Rebuttal · Authors · 2026-03-27
>
> We thank the reviewer for the careful reading of our paper.
>
> On the broader question of **advantage over prior work**: our contribution is not a better regret rate, nor do we claim a clear empirical or runtime advantage over the strongest existing baselines. The contribution is conceptual, and we believe it addresses an important gap in the literature. In RL, and specifically in CMABs, there is currently a marked separation between the theoretical literature and the empirical literature: on the theory side, the known optimal CMAB algorithms are based on deterministic optimism or inverse-gap-weighted policies, whereas in practice one typically runs variants of policy optimization methods, such as PPO, GRPO, and related approaches. Thus, unlike simpler settings such as bandits, where EXP3 already gives a policy-optimiztion based theoretical framework, or tabular MDPs, where optimistic policy optimization is by now well understood, for CMAB with general function approximation there has been no comparable theoretical understanding of policy optimization. In this sense, the absence of a policy-optimization theory for CMAB is substantial in literature. Our work is, to our knowledge, the first to study pure KL-regularized policy-optimization update for this CMAB model and prove an optimal high-probability regret, thereby closing an important gap between theory and practice.
>
> We view this as a meaningful conceptual contribution to the RL literature, not only because it clarifies the statistical power of policy optimization in this setting, but also because it opens the way to many natural future directions, including more practical scalable variants and richer RL models.We will revise the framing to make this point more precise and to more clearly acknowledge that the current $O(K^2)$ implementation is mainly a proof-of-concept implementation rather than a claim of practical superiority; indeed, the paper already notes this implementation overhead and presents the experiments primarily as evidence of competitiveness rather than dominance .
>
> Regarding **infinite function classes**: the role of Corollary 4.1 is simply that of a uniform convergence guarantee for the ERM outputs. In the current version, we instantiate it in the finite-class setting, which yields the ($\log |F|$) term. More generally, the same step extends in the standard way by replacing this finite-class term with any appropriate learning complexity measure that guarantees uniform convergence for ERM. For example, one may use VC dimension for binary rewards, fat-shattering dimension for continuous rewards, and more refined notions such as the Eluder dimension in richer function-approximation settings. In each case, Corollary 4.1 is replaced by the corresponding uniform convergence bound for ERM, and the regret bound changes accordingly by substituting ($\log |F|$) with the relevant complexity measure. Thus, the finite-class statement in the paper should be viewed as the simplest instantiation of a more general uniform-convergence ingredient, rather than as something inherently tied to finite classes.
>
> Regarding **Lemma A.2**: this step uses a standard fresh-context conditioning argument in stochastic CMAB analyses; see, for example, the proof of Lemma 1 in Xu and Zeevi (2020). In more detail, Lemma B.4 holds uniformly for any fixed sequence of functions $f_2,f_3,\ldots \in F$. Corollary 4.1 is simply the specialization of Lemma B.4 to the particular fixed sequence given by the square-loss minimizers. Thus, in the proof, we are allowed to view the sequence $\hat f_2,\ldots,\hat f_k$ of minimizers as fixed when invoking Corollary 4.1. Separately, conditioning on the history $H_{i-1}$ for each $i<k$ is only used to fix the policy $\pi_i$. Once $\pi_i$ is fixed, and the minimizer sequence is treated as fixed through the uniformity of Lemma B.4, the remaining quantity is simply a deterministic function of the context argument.
> We then evaluate this function under the context distribution $D$, so the symbol used for the context variable is immaterial: it may be written as $c_i$, $c_k$, or $c$, provided it is understood as any context distributed according to $D$, rather than as a realized past sample. Since the contexts are i.i.d., these variables play the same role in the lemma as generic draws from $D$. This is why the displayed equalities and inequalities are valid. In the revision, we will make this conditioning step explicit in the proof so that the argument is completely transparent.
>
> Regarding **explicitly stating the $\eta,\beta$** assignments, we will definitely add this in the revised version.
>
> We hope our responses clarify your concerns and if the reviewer is convinced by the above clarifications, we hope they will consider raising their evaluation accordingly. Otherwise, we will be happy to engage further during the discussion period.

---

> > ### Author Rebuttal · Reviewer_TPPa · 2026-04-01
> >
> > The rebuttal addresses my concerns regarding Lemma A.2 and I add 1 point

---

### Official Review · Reviewer_8wcV · 2026-03-13

**Soundness:** 3
**Presentation:** 3
**Significance:** 2
**Originality:** 3
**Overall Recommendation:** 4
**Confidence:** 3

**Summary:**

This paper studies stochastic contextual bandits with realizable offline function approximation and proposes OPO-CMAB, a policy-optimization-style algorithm that combines least-squares loss prediction, a counterfactual exploration bonus for stochastic policies, and exponential / mirror-descent updates. The main result is a high-probability optimal regret bound in the standard regression-oracle setting. The paper is generally clear and technically well organized. However, I am not fully convinced by the claimed conceptual novelty: many of the main ingredients appear closely related to prior counterfactual-bonus methods in CMAB and prior optimistic policy-optimization analyses in MDPs.

**Compliance With Llm Reviewing Policy:**

Affirmed.

**Final Justification:**

My concerns have been adequately addressed. I have increased my score.

**Key Questions For Authors:**

- Could the authors clarify more precisely what is genuinely new beyond adapting prior counterfactual bonus ideas to stochastic policies and reusing an existing optimistic PO proof template?
- Since policy-based exploration already has strong prior theory in several MDP settings, why should the compatibility of policy optimization with CMAB be viewed as a strong conceptual insight, rather than mainly a setting-specific theorem completion?

**Limitations:**

yes

**Strengths And Weaknesses:**

Strengths
- The paper fills a specific gap in the theory landscape by giving a pure policy-optimization algorithm with a high-probability optimal regret guarantee in the standard stochastic CMAB + offline regression-oracle setting.
- The analysis is fairly clean and modular, and the stochastic-policy version of the counterfactual exploration bonus is a reasonable technical adaptation.

Weaknesses
- The conceptual novelty seems limited. At a high level, the paper appears to combine a Xu--Zeevi-style counterfactual optimism mechanism with a Shani/optimistic-PO-style analysis, rather than introducing a substantially new idea.
- The empirical section is not especially compelling: the reported performance is broadly similar to prior baselines, and the current implementation has nontrivial overhead, which makes the practical benefit of the method unclear.

---

> ### Author Rebuttal · Authors · 2026-03-27
>
> We thank the reviewer for the careful reading of our paper
> We agree that our work builds on important prior ingredients, and we do not claim to develop an entirely disconnected paradigm. Rather, our claim is that the genuine novelty lies in showing that these ingredients can be combined in a nontrivial way to obtain the first pure policy-optimization algorithm with optimal high-probability regret in stochastic CMAB with general offline function approximation.
>  More broadly, our motivation for studying this problem is to address an important gap between theory and practice: in CMABs, the algorithms with the strongest theoretical guarantees are often structurally different from the policy-optimization methods that are most commonly used in practice (i.e., PPO, GRPO, etc). In this sense, we hope the reviewer can also assess the contribution through the technique behind the optimal regret bound.
> More precisely, the novelty is not merely “taking Xu–Zeevi bonuses and plugging them into Shani et al.” First, Xu and Zeevi’s counterfactual bonuses were developed for deterministic policies; extending them to stochastic policies is not cosmetic, because the relevant exploration quantity becomes the cumulative policy mass $\sum_{i<k}\pi_i(c,a)$, and the regret analysis must control this quantity in a way that is compatible with a fully stochastic mirror-descent update.
> Second, optimistic policy-optimization analyses in MDPs rely on finite small state and action spaces that are fundamentally different from contextual bandits with rich context space, where contexts typically do not repeat. Hence, tabular approximation is infeasible and function approximation adds its unavoidable additional complications.
> Thus, transferring the optimistic policy-optimization viewpoint to CMAB is not immediate. one must introduce a new counterfactual notion of exploration that substitutes for visitation counts and is analyzable under regression-based loss prediction. Third, our proof is not just a reuse of an existing template, but a new integration of three components that had not previously been shown compatible in this setting: (i) offline least-squares regression convergence guarantees, (ii) counterfactual optimism (iii) KL-regularized policy improvement. This combination is exactly what yields the optimal $\tilde O(\sqrt{K|A|\log|F|})$ regret in a genuinely policy-optimization-based algorithm. We will revise the paper to state this contribution more precisely.
>
> Regarding the second question, we strongly disagree that one can view our result as a “theorem completion,” as our result is far from a trivial adaptation of any known algorithm. Policy-based exploration already has strong theory in several MDP settings, but CMAB is the natural minimal setting where one can ask whether practical policy-optimization ideas remain compatible with optimal statistical guarantees outside tabular or linear MDP structure. In that sense, the compatibility is conceptually meaningful for two reasons. First, policy optimization is arguably the most practically influential paradigm in modern RL and CMAB, yet the theoretical optimal-regret CMAB literature has so far been dominated by structurally different approaches such as UCB-style optimism or inverse-gap weighting. Showing that KL-regularized policy optimization can also achieve the same optimal guarantee closes an important conceptual gap between what is used in practice and what is known in theory. Second, this compatibility is not automatic: in tabular MDPs, exploration is naturally tied to state-action visitation, whereas in CMAB with large context spaces one must reason counterfactually across contexts and establish that the policy update still induces enough exploration when combined with regression-based prediction. Thus, even if one views the result as completing the picture for CMABs, it is a nontrivial completion that shows the robustness of the policy-optimization principle beyond tabular and linear MDPs. We therefore hope the reviewer can also see the impact of the proof technique behind the regret bound: beyond establishing optimality, it shows that policy optimization can be analyzed in a rich CMAB setting using counterfactual optimistic bonuses, which may be useful for future extensions to more  general settings.
>
> On the empirical side, we agree that our current implementation does not demonstrate a practical advantage over the baselines, and we will revise the framing to make clear that the empirical goal is to show competitiveness and applicability rather than superiority.
> Our main contribution is theoretical: to establish that pure policy optimization is possible in this standard CMAB model while retaining optimal regret, and we see scalable implementations and heuristics as an important next step.
>
>  In light of the strengths already identified by the reviewe, and if they convinced by the above clarifications, we hope they will consider raising their evaluation accordingly.

---

> > ### Author Rebuttal · Reviewer_8wcV · 2026-04-04
> >
> > I thank the authors for their rebuttal. My concerns have been adequately addressed. I have increased my score.

---

### Official Review · Reviewer_A5vG · 2026-03-14

**Soundness:** 3
**Presentation:** 3
**Significance:** 2
**Originality:** 3
**Overall Recommendation:** 4
**Confidence:** 3

**Summary:**

This paper proposes OPO-CMAB, a policy-optimization-based algorithm for stochastic CMAB, by bringing into the CMAB setting the policy optimization approach that has recently attracted substantial attention in reinforcement learning. The paper shows that OPO-CMAB achieves a regret bound comparable to those of existing regression-oracle-based methods, such as FALCON by Simchi-Levi and Xu (2022).

**Compliance With Llm Reviewing Policy:**

Affirmed.

**Final Justification:**

Final justification:

The rebuttal clarified several important points. In particular, the authors now explain more clearly that by “policy optimization” they specifically mean an OMD / KL-regularized policy-improvement framework, and their discussion of the stochastic-context assumption makes the role of the counterfactual bonus substantially clearer. These clarifications improve the paper’s positioning and help justify some of the modeling choices.

That said, my overall assessment remains unchanged. While I agree that it is conceptually interesting to show that a pure KL-regularized policy-optimization approach can achieve optimal regret in stochastic CMAB with function approximation, I still find the contribution somewhat nuanced relative to prior work such as Simchi-Levi and Xu (2022), which studies a very similar setting and achieves comparable guarantees through a structurally different method. In my view, the rebuttal strengthens the interpretation of the paper as a conceptual and methodological contribution, but it does not fully establish a sufficiently compelling practical or algorithmic advantage over existing optimal approaches.

Overall, I consider my concerns partially resolved, but not enough to revise my score. The paper is technically solid and reasonably well presented, but my evaluation remains at borderline accept.

**Key Questions For Authors:**

1. The paper would benefit from a more detailed explanation of why the stochasticity assumption on the context is necessary. Which part of the analysis or algorithm specifically makes this restriction unavoidable?

2. The authors should provide a clearer explanation of the policy optimization framework they want to emphasize.

3. It would also be helpful to explain why the counterfactual approach is necessary here, and in what sense an optimism-based approach would make the analysis more difficult.

**Strengths And Weaknesses:**

First, as the authors themselves also note, the paper strongly reminds me of Simchi-Levi and Xu (2022), which studies a very similar problem setting, including the assumption of a regression oracle. Both works learn using an offline, square-loss-based regression oracle, and both achieve regret on the order of (\sqrt{K|A| \log |F|}), so the comparison is unavoidable. Indeed, I found the results to be substantially similar in many respects.

Of course, the two algorithms are structurally quite different. Unlike the present paper, which is analyzed within the framework of a fairly standard exponential-weight-based policy optimization approach, algorithms such as FALCON choose a rather particular probability weighting corresponding to a best-versus-rest-of-arm structure. In that sense, it is certainly interesting that two fundamentally different approaches arrive at comparable results.

That said, adding diversity within an already established line of work makes the significance somewhat nuanced. It is clearly a contribution, but I think the paper is somewhat less convincing in arguing why this approach is attractive enough to replace existing algorithms. Moreover, the requirement that the context must be stochastic also appears to be a restriction introduced by adopting the policy optimization approach.

In addition, the positioning of the paper itself feels somewhat ambiguous. As is well known, “policy optimization” in reinforcement learning is not a single monolithic methodology. In fact, although this paper is inspired by Shani et al. (2020), it looks rather different in important ways: Shani et al. clearly assign optimistic (Q)-values, whereas this paper uses a counterfactual exploration bonus. Nevertheless, the paper does not clearly articulate what precise framework of “policy optimization” it has in mind. As a result, the reader is left to fill in by intuition what it means for the paper to have faithfully translated the concept of policy optimization into the CMAB setting. I would strongly encourage the authors to make this point explicit, even if only in the appendix.

I have focused mostly on criticisms so far, but I should also note that the clarity of the proof techniques appears to be relatively strong. I did not have time to read every detail completely, but the paper seems to explain in detail how it adapts and twists existing proof flows. The proof strategy itself also felt fresh and distinct from those used in FALCON, SquareCB, or FastCB, which is of course natural given its policy-optimization-based perspective. The references also seem to cover the relevant related works reasonably well, and the authors appear to have made a genuine effort to highlight the distinctions of their paper.

Taking all of this into account, I have decided to give the paper a borderline accept. However, I want to emphasize that this decision is very close to the boundary.

---

> ### Author Rebuttal · Authors · 2026-03-27
>
> We thank the reviewer for the thoughtful feedback.
>
> We agree that our setting is the same as Simchi-Levi and Xu (2022); indeed, this is the standard stochastic CMAB setup used by most prior regret for CMAB works. However, our motivation for revisiting this well-established setting is different: while the existing optimal algorithms are based on deterministic policy or inverse-gap-weighting rules, in practice, CMAB systems are often trained using policy optimization, most notably as PPO-style methods. More broadly, we view this as an important gap in the literature: in stochastic CMAB with general function approximation, there has so far been no comparable theoretical understanding of pure policy-optimization methods, despite their central role in practical RL and CMAB systems in particular.
>  At the high level, PPO is built around minimizing an expected loss together with a KL-regularization term, and this is exactly the policy optimization framework we study here: at each round, we update the policy by minimizing the estimated expected loss plus a KL penalty relative to the previous policy, which in the bandit setting yields the familiar exponential-weights / EXP3 / OMD-style update. Thus, the point of our paper is to provide the first theoretically provable optimal version of this practical policy-optimization paradigm for stochastic CMAB with general function approximation; thereby justifying why such methods are appealing in practice.
> In this sense, we hope the reviewer can also view the significance of the result through the technique; the contribution is conceptual, in showing that a pure KL-regularized policy-optimization approach can in fact be made elegantly optimal in CMABs.
>
> To directly address the reviewer’s question about what notion of “policy optimization” we mean, we refer specifically to OMD-style update, i.e., KL-regularized policy improvement, which minimizing the current estimated expected loss plus a KL-divergence penalty to the previous policy. In bandits, this reduces to exponential policy updates with respect to the estimated loss. We agree this should be stated explicitly in the main paper (we refer the reviewer to appendix B.1. for explanation of OMD), and in the revision we will define this framework clearly and connect it both to OMD updates.
>
> Regarding the stochastic-context assumption, this is not an incidental restriction but is tied directly to our bonus construction and analysis. Our counterfactual bonuses depend on the quantity $\sum_{i<k}\pi_i(c,a)$, which represents how often arm $a$ would have been selected by the past policies had the current context $c$ appeared previously. This quantity is meaningful precisely because the contexts are i.i.d.: under stochastic contexts, the current context can be viewed as sampled from the same distribution as past contexts, so these counterfactual visitation probabilities can be related to the actual learning process. Informally, this is the sense in which one may think of the algorithm as reasoning about a “mean” or representative context distribution. If contexts were adversarial, this quantity would no longer be connected to the observed data in a controlled way, and the key step in the analysis, using these counterfactual counts to quantify exploration and relate them to regression error, would break down.
>
> Finally, on why the counterfactual approach is needed and how it relates to optimism: our method is still optimistic in spirit. As in optimistic policy optimization for RL, we construct optimistic estimates by subtracting bonuses from the estimated losses and then update the policy using these optimistic values. The main difference is that in CMAB with rich context space we cannot define exploration bonuses using actual action visitation counts as in tabular RL or stochastic MAB. Instead, simce a context may never repeat, we need a counterfactual notion of exploration: how often each action would have been played under the current context according to the history of past policies. This is why the counterfactual bonus is necessary. In that sense, our method adapting optimistic policy optimization to the stochastic CMAB setting.
> We will revise the paper to make this point clearer and to better emphasize that the novelty lies in showing that this policy-optimization viewpoint can achieve the same optimal regret guarantees previously known only for structurally very different CMAB algorithms. We also hope to make clearer that the impact of the result is not only the regret guarantee itself, but the proof technique showing that policy optimization admits an optimal regret in this model, which may be useful for future extensions to more practical and richer settings.
>
> We hope we have addressed your questions and concerns, and we would be happy to engage further during the discussion if helpful. We also hope we have conveyed the significance of this work to the RL community, and that you will consider revisiting your opinion.

---

> > ### Author Rebuttal · Reviewer_A5vG · 2026-04-05
> >
> > The rebuttal addresses several of my concerns, and I appreciate that the authors now articulate more clearly what they mean by “policy optimization,” namely an OMD / KL-regularized policy-improvement view. I also found their explanation of why the stochastic-context assumption is tied to the counterfactual bonus construction helpful; this makes the restriction feel more principled than incidental.
> >
> > That said, I still view the contribution as somewhat nuanced. The rebuttal improves the paper’s positioning, but it does not fully remove my concern that the practical and conceptual advantage over prior optimal CMAB methods remains only partially established. In particular, the main value seems to lie in showing that a pure KL-regularized policy-optimization approach can also attain optimal regret, rather than in demonstrating a clearly stronger algorithmic consequence over existing methods.
> >
> > Overall, I consider my concerns partially resolved, but not enough to change my overall assessment. I will keep my score unchanged.

---

### Decision · Program_Chairs · 2026-04-30

**Decision:**

Accept (regular)

**Comment:**

This paper provides a first analysis for policy optimization for realizability contextual bandits with a given function set and offline regression oracle.  The setting is same as FALCON and UCCB, but the algorithm is of a different type.  They prove the same bounds as prior work.

As pointed out in multiple reviews, there is no convincing argument on why this new algorithm is more favorable than FALCON or UCCB in terms of regret or computational.  However, I agree that the contribution is in showing a practically popular approach works in theory. It better complete the picture of the CMAB problem. I therefore recommend acceptance.